# On the cohesion and separability of average-link for hierarchical agglomerative clustering

**Eduardo S. Laber**
Departmento de Informática, PUC-RIO
laber@inf.puc-rio.br

**Miguel Batista**
Departmento de Informática, PUC-RIO
miguel260503@gmail.com

## Abstract

Average-link is widely recognized as one of the most popular and effective methods for building hierarchical agglomerative clustering. The available theoretical analyses show that this method has a much better approximation than other popular heuristics, as single-linkage and complete-linkage, regarding variants of Dasgupta's cost function [STOC 2016]. However, these analyses do not separate average-link from a random hierarchy and they are not appealing for metric spaces since every hierarchical clustering has a $1/2$ approximation with regard to the variant of Dasgupta's function that is employed for dissimilarity measures [Moseley and Yang 2020]. In this paper, we present a comprehensive study of the performance of `average-link` in metric spaces, regarding several natural criteria that capture separability and cohesion, and are more interpretable than Dasgupta's cost function and its variants. We also present experimental results with real datasets that, together with our theoretical analyses, suggest that average-link is a better choice than other related methods when both cohesion and separability are important goals.

## 1 Introduction

Clustering is the task of partitioning a set of objects/points so that similar ones are grouped together while dissimilar ones are put in different groups. Clustering methods are widely used for exploratory analysis and for reducing the computational resources required to handle large datasets.

Hierarchical clustering is an important class of clustering methods. Given a set of $\mathcal{X}$ of $n$ points, a hierarchical clustering is a sequence of clusterings $(\mathcal{C}^n, \mathcal{C}^{n-1}, \dots, \mathcal{C}^1)$, where $\mathcal{C}^n$ is a clustering with $n$ unitary clusters, each of them corresponding to a point in $\mathcal{X}$, and the clustering $\mathcal{C}^i$, with $i < n$, is obtained from $\mathcal{C}^{i+1}$ by replacing two of its clusters with their union $A^i$. A hierarchical clustering induces a strictly binary tree with $n$ leaves, where each leaf corresponds to a point in $\mathcal{X}$ and the $i$th internal node, with $i < n$, is associated with the cluster $A^i$; the points in $A^i$ correspond to the leaves of the subtree rooted in $A^i$. Hierarchical clustering methods are often taught in data science/ML courses, are implemented in many machine learning libraries, such as `scipy`, and have applications in different fields as evolution studies via phylogenetic trees [Eisen et al., 1998], finance [Tumminello et al., 2010] and detection of closely related entities [Kobren et al., 2017, Monath et al., 2021].

`Average-link` is widely considered one of the most effective hierarchical clustering algorithms. It belongs to the class of *agglomerative methods*, that is, methods that start with a set of $n$ clusters, corresponding to the $n$ input points, and iteratively use a linkage rule to merge two clusters. Due to its relevance, we can find some recent works dedicated to improving `average-link`' efficiency and scalability [Yu et al., 2021, Dhulipala et al., 2021, 2022, 2023] as well as recent theoretical work that try to understand its success in practice [Cohen-Addad et al., 2019, Charikar et al., 2019a, Moseley and Wang, 2023, Charikar et al., 2019b].

38th Conference on Neural Information Processing Systems (NeurIPS 2024).

Most of the available theoretical works give approximation bounds for `average-link` regarding the cost function introduced by [Dasgupta, 2016] as well as for some variants of it. Let $\mathcal{D}$ be the tree induced by a hierarchical clustering. Dasgupta's cost function and its variation for dissimilarities considered in [Cohen-Addad et al., 2019] are, respectively, given by

$$\texttt{Dasg}(\mathcal{D}) = \sum_{a,b \in \mathcal{X}} \texttt{sim}(a,b) \cdot |D(a,b)| \ \text{ and } \ \texttt{CKMM}(\mathcal{D}) = \sum_{a,b \in \mathcal{X}} \texttt{diss}(a,b) \cdot |D(a,b)|, \quad (1)$$

where $\texttt{sim}(a,b)$ ($\texttt{diss}(a,b)$) is the similarity (dissimilarity) of points $a$ and $b$; $D(a,b)$ is the subtree of $\mathcal{D}$ rooted at the least common ancestor of the leaves corresponding to $a$ and $b$, and $|D(a,b)|$ is the number of leaves in $D(a,b)$. In general, the existing results show that `average-link` achieves constant approximation for variants of Dasgupta's function while other linkage methods do not.

However, there is significant room for further analysis due to the following reasons. First, Dasgupta's cost function, despite its nice properties, is less interpretable than traditional cost functions that measure compactness and separability. Second, although the analyses based on `Dasg` and its variants allow to separate `average-link` from other linkage methods as `single-linkage` and `complete-linkage` in terms of approximation, they do not separate `average-link` from a random hierarchy [Cohen-Addad et al., 2019, Moseley and Wang, 2023, Charikar et al., 2019b]. Moreover, for the case in which the points lie in a metric space every hierarchical clustering has $1/2$ approximation for the maximization of `CKMM` [Wang and Moseley, 2020], so this cost function is less appealing in this relevant setting. Finally, to the best of our knowledge, `Dasg` does not reveal how good are the clusters generated for a specific range of $k$. As an example, small $k$ are important for exploratory analysis while large $k$ is important for de-duplication tasks [Kobren et al., 2017].

## 1.1 Our results

Motivated by this scenario, we present a comprehensive study of the performance of `average-link` in metric spaces, with regards to several natural criteria that capture separability and cohesion of clustering. In a nutshell, these results, as explained below, show that average link has much better global properties than other popular heuristics when these two important goals are taken into account.

Let $(\mathcal{X}, \texttt{dist})$ be a metric space, where $\mathcal{X}$ is a set of $n$ points. The diameter $\texttt{diam}(S)$ of a set of points $S$ is given by $\texttt{diam}(S) = \max\{\texttt{dist}(x,y) | x, y \in S\}$. For a cluster $A$ and for two clusters $A$ and $B$, let

$$\texttt{avg}(A) = \frac{1}{\binom{|A|}{2}} \sum_{x,y \in A} \texttt{dist}(x,y) \ \text{ and } \ \texttt{avg}(A,B) = \frac{1}{|A| \cdot |B|} \sum_{x \in A} \sum_{y \in B} \texttt{dist}(x,y)$$

Let $\mathcal{C} = (C_1, \ldots, C_k)$ be a $k$-clustering for $(\mathcal{X}, \texttt{dist})$. To study separability we consider the average ($\texttt{sep}_{\texttt{av}}$) and the minimum ($\texttt{sep}_{\texttt{min}}$) avg among clusters in $\mathcal{C}$, that is,

$$\texttt{sep}_{\texttt{av}}(\mathcal{C}) := \frac{1}{\binom{k}{2}} \sum_{i \neq j} \texttt{avg}(C_i, C_j) \ \text{ and } \ \texttt{sep}_{\texttt{min}}(\mathcal{C}) := \min_{i \neq j}\{\texttt{avg}(C_i, C_j)\}, \quad (2)$$

On the other hand, for studying cohesion, we consider the maximum diameter (`max-diam`) and the maximum average pairwise distance (`max-avg`) of the clusters in $\mathcal{C}$. In formulae,

$$\texttt{max-diam}(\mathcal{C}) := \max\{\texttt{diam}(C_i) | 1 \leq i \leq k\} \ \text{ and } \ \texttt{max-avg}(\mathcal{C}) := \max\{\texttt{avg}(C_i) | 1 \leq i \leq k\} \tag{3}$$

We also study natural optimization goals that capture both the separability and the cohesion of a clustering. We define the `cs-ratio`$_{\texttt{AV}}$ and `cs-ratio`$_{\texttt{DM}}$ of a clustering $\mathcal{C}$ as

$$\texttt{cs-ratio}_{\texttt{AV}}(\mathcal{C}) := \frac{\texttt{max-avg}(\mathcal{C})}{\texttt{sep}_{\texttt{min}}(\mathcal{C})} \ \text{ and } \ \texttt{cs-ratio}_{\texttt{DM}}(\mathcal{C}) := \frac{\texttt{max-diam}(\mathcal{C})}{\texttt{sep}_{\texttt{min}}(\mathcal{C})} \tag{4}$$

Let $\mathcal{A}^k$ be a $k$-clustering produced by `average-link`. We first prove through a simple inductive argument that $\texttt{cs-ratio}_{\texttt{AV}}(\mathcal{A}^k) \leq 1$. This result does not assume that the points in $\mathcal{X}$ lie in a metric space and it is tight in the sense that there are instances in which $\texttt{cs-ratio}_{\texttt{AV}}(\mathcal{C}) = 1$ for every

$k$-clustering $\mathcal{C}$. For the related `cs-ratio`$_\text{DM}$ criterion, we present a more involved analysis which shows that `cs-ratio`$_\text{DM}(\mathcal{A}^k)$ as well as the approximation of `average-link` regarding OPT (the minimum possible `cs-ratio`$_\text{DM}$) are $O(\log n)$; these bounds are nearly tight since there exists an instance for which `cs-ratio`$_\text{DM}(\mathcal{A}^k)$ and `cs-ratio`$_\text{DM}(\mathcal{A}^k)/$OPT are $\Omega(\frac{\log n}{\log \log n})$. Both `cs-ratio`$_\text{AV}$ and `cs-ratio`$_\text{DM}$ allow an exponential separation between `average-link` and other linkage methods, as `single-linkage` and `complete-linkage`. Interestingly, in contrast to `CKMM` (Eq. 1), our criteria also allow a very clear separation between `average-link` and the clustering induced by a random hierarchy.

Next, we focus on separability criteria. Let $\text{OPT}_\text{SEP}(k)$ be the maximum possible `sep`$_\text{av}$ of a $k$-clustering for $(\mathcal{X}, \texttt{dist})$. We show that `sep`$_\text{av}(\mathcal{A}^k)$ is at least $\frac{\text{OPT}_\text{SEP}(k)}{k+2 \ln n}$ and that this result is nearly tight. Furthermore, we argue that any hierarchical clustering algorithm that has bounded approximation regarding `max-diam` or `max-avg` does not have approximation better than $1/k$ to `sep`$_\text{av}$. Regarding `single-linkage` and `complete-linkage`, we present instances that show that their approximation with respect to `sep`$_\text{av}$ are exponentially worse than that of `average-link`, for the relevant case that $k$ is small.

We also investigate the cohesion of `average-link`. For a $k$-clustering $\mathcal{C}$, let `avg-diam` be the average diameter of the $k$ clusters in $\mathcal{C}$. Let $\text{OPT}_\text{DM}(k)$ and $\text{OPT}_\text{AV}(k)$ be, respectively, the minimum possible `max-diam` and `avg-diam` of a $k$-clustering for $(\mathcal{X}, \texttt{dist})$. We prove that for all $k$, `max-diam`$(\mathcal{A}^k) \leq \min\{k, 1 + 4 \ln n\}k^{\log_2 3}\text{OPT}_\text{AV}(k)$. This result together with the instance given by Theorem 3.4 of [Dasgupta and Laber, 2024] allow to separate `average-link` from `single-linkage`, in terms of approximation, when $k$ is $\Omega(\log^{2.41} n)$. We also show that `max-diam`$(\mathcal{A}^k)$ is $\Omega(k)\text{OPT}_\text{DM}(k)$, which is, to the best of our knowledge, the first lower bound on the maximum diameter of `average-link`.

Finally, to **complement** our study, we present some experiments with 10 real datasets in which we evaluate, to some extent, if our theoretical results line up with what is observed in practice. These experiments conform with our theoretical results since they also suggest that `average-link` performs better than other related methods when both cohesion and separability are taken into account.

## 1.2 Related work

There is a vast literature about hierarchical agglomerative clustering methods. Here, we focus on works that provide provable guarantees for `average-link` and some other well-known linkage methods.

**Average-link**. There are works that present bounds on the approximation of `average-link` regarding some criteria [Cohen-Addad et al., 2019, Charikar et al., 2019b,a, Moseley and Wang, 2023, Dasgupta and Laber, 2024]. All these works but [Dasgupta and Laber, 2024] analyse the approximation of `average-link` regarding variants of Dasgupta's cost function. [Moseley and Wang, 2023] assumes that the proximity between the points in $\mathcal{X}$ is given by a similarity matrix. They show that `average-link` is a $1/3$-approximation with respect to the "dual" of Dasgupta's cost function. [Cohen-Addad et al., 2019], as in our work, assumes that the proximity between points in $\mathcal{X}$ is given by a dissimilarity measure and shows that `average-link` has $2/3$ approximation for the problem of maximizing `CKMM` (Eq. 1). [Charikar et al., 2019b] show that these approximation ratio for `average-link` are tight. These papers also show that a random hierarchy obtained by a divisive heuristic that randomly splits the set of points in each cluster matches the $1/3$ and $2/3$ bounds.

[Dasgupta and Laber, 2024] presents an interesting approach to derive upper bounds on cohesion criteria for a certain class of linkage methods that includes `average-link`. They show that $\texttt{avg}(A) \leq k^{1.59}\text{OPT}_\text{AV}(k)$ for every cluster $A \in \mathcal{A}^k$. Our bound on the maximum diameter of a cluster in $\mathcal{A}^k$ incurs an extra factor of $\min\{k, 1 + 4 \ln n\}$ to this bound and its proof combines their approach with some new ideas/analyses.

**Other Linkage Methods**. There are also works that give bounds on the diameter of the clustering built by `complete-linkage` and `single-linkage` on metric spaces [Dasgupta and Long, 2005, Ackermann et al., 2010, Großwendt and Röglin, 2015, Arutyunova et al., 2023, Dasgupta and Laber, 2024]. Let $\mathcal{C}$ and $\mathcal{S}$ be the $k$-clustering built by these methods, respectively. [Arutyunova et al., 2023] shows that `max-diam`$(\mathcal{C})$ is $\Omega(k\text{OPT}_\text{DM}(k))$ while [Dasgupta

and Laber, 2024] shows that `max-diam`$(\mathcal{C})$ is $O(\min\{k^{1.30}\text{OPT}_{\text{DM}}(k), k^{1.59}\text{OPT}_{\text{AV}}(k)\})$. Regarding `single-linkage`, `max-diam`$(\mathcal{S})$ is $\Theta(k\text{OPT}_{\text{DM}}(k))$ [Dasgupta and Long, 2005, Arutyunova et al., 2023] and $\Omega(k^2\text{OPT}_{\text{AV}}(k))$ [Dasgupta and Laber, 2024]. [Ackermann et al., 2010, Großwendt and Röglin, 2015] give bounds for the case in which `dist` is the Euclidean metric.

In terms of separability criteria, it is well known that `single-linkage` maximizes the minimum spacing of a clustering [Kleinberg and Tardos, 2006][Chap 4.7]. Recently, [Laber and Murtinho, 2023] observed that it also maximizes the cost of the minimum spanning tree spacing, a stronger criterion. These criteria, in contrast to ours, just take into account the minimum distance between points in different clusters and then they can be significantly impacted by noise.

[Großwendt et al., 2019] shows that Ward's method gives a 2-approximation for $k$-means when the optimal clusters are well-separated.

## 2 Preliminaries

Algorithm 2 shows a pseudo-code for `average-link`. The function $\text{dist}_{AL}(A, B)$ at line 3 that measures the distance between clusters $A$ and $B$ is given by

$$\text{dist}_{AL}(A, B) := \frac{1}{|A||B|} \sum_{a \in A} \sum_{b \in B} \text{dist}(a, b).$$

`single-linkage` and `complete-linkage` are obtained by replacing $\text{dist}_{AL}$, in Algorithm 2, with $\text{dist}_{SL}(A, B) := \min\{\text{dist}(a, b)|(a, b) \in A \times B\}$ and $\text{dist}_{CL}(A, B) := \max\{\text{dist}(a, b)|(a, b) \in A \times B\}$, respectively.

---

**Algorithm 2** Average Link

1: $\mathcal{A}^n \leftarrow$ clustering with $n$ unitary clusters, each one containing a point of $\mathcal{X}$
2: **For** $i = n - 1$ down to 1
3:    $(A, B) \leftarrow$ clusters in $\mathcal{A}^{i+1}$ for which $\text{dist}_{AL}(A, B)$ is minimum
4:    $\mathcal{A}^i \leftarrow \mathcal{A}^{i+1} - \{A\} - \{B\} \cup \{A \cup B\}$

---

A version of the triangle inequality for averages will be employed a number of times in our analyses. Its proof can be found in Section A.

**Proposition 2.1** (Triangle Inequality for averages). *Let $A$, $B$ and $C$ be three clusters. Then,*

$$\text{avg}(A, C) \leq \text{avg}(A, B) + \text{avg}(B, C).$$

For two disjoint clusters $A$ and $B$, the following identity holds

$$\binom{(|A| + |B|)}{2} \text{avg}(A \cup B) = \binom{|A|}{2} \text{avg}(A) + |A||B|\text{avg}(A, B) + \binom{|B|}{2} \text{avg}(B).$$

Dividing both sides by $\binom{(|A|+|B|)}{2}$, we conclude that $\text{avg}(A \cup B)$ is a convex combination of $\text{avg}(A), \text{avg}(B)$ and $\text{avg}(A, B)$, a fact will be used a couple of times in our analyses.

The following notation will be used throughout the text. We use $H_p = \sum_{i=1}^p \frac{1}{i}$ to denote the $p$th harmonic number and $\mathcal{A}^k$ to refer to the $k$-clustering obtained by `average-link` for the instance under consideration, which will always be clear from the context.

## 3 Cohesion and separability

In this section, we analyze the performance of `average-link` with respect to both `cs-ratio`$_{\text{AV}}$ and `cs-ratio`$_{\text{DM}}$ (Eq. 4), criteria that simultaneously take into account the separability and the cohesion of a clustering. Moreover, we contrast its performance with that achieved by other linkage methods.

## 3.1 The `cs-ratio`$_{\text{AV}}$ criterion

We first show that `cs-ratio`$_{\text{AV}}(\mathcal{A}^k) \leq 1$. The proof of this result can be found in Section B.1, it uses induction on the number of iterations of `average-link` together with a fairly simple case analysis.

**Theorem 3.1.** *Let* $\mathcal{A}^k$ *be a* $k$-*clustering built by* `average-link`. *Then, for every* $k$, `cs-ratio`$_{\text{AV}}(\mathcal{A}^k) \leq 1$.

We note that the above result does not assume the triangle inequality and it is tight in the sense that for the instance $(\mathcal{X}, \text{dist})$, in which the $n$ points of $\mathcal{X}$ have pairwise distance 1, every clustering has `cs-ratio`$_{\text{AV}}$ equal to 1.

In Section B.2, we present instances which show that `cs-ratio`$_{\text{AV}}$ can be $\Omega(n)$, $\Omega(\sqrt{n})$ and unbounded in terms of $n$ for `single-linkage`, `complete-linkage` and a random hierarchy, respectively. Interestingly, all the $k$-clustering, with $2 < k \leq n/2$, induced by the hierarchical clustering obtained by these methods satisfy these bounds. Furthermore, since `cs-ratio`$_{\text{DM}}(\mathcal{C}) \geq$ `cs-ratio`$_{\text{AV}}(\mathcal{C})$ for every clustering $\mathcal{C}$, these bounds also hold for the `cs-ratio`$_{\text{DM}}$ criterion.

A natural question that arises is whether `average-link` has a "good" approximation with respect to `cs-ratio`$_{\text{AV}}$. Unfortunately, the answer is no. In fact, in Section B.3 we show an instance where the approximation is unbounded in terms of $n$. However, as we show in the next section, `average-link` has a logarithmic approximation with respect to `cs-ratio`$_{\text{DM}}$.

## 3.2 The `cs-ratio`$_{\text{DM}}$ criterion

We analyze the `cs-ratio`$_{\text{DM}}$ of `average-link`. The results of this section will have an important role in the analysis of both the separability and cohesion of `average-link` presented further.

First, we show that for every cluster $X$ in $\mathcal{A}^k$, the average distance of a point $x \in X$ to the other points in $X - x$ is at most a logarithmic factor of the average distance between any two clusters $Y$ and $Z$. The proof can be found in Section B.5. Let $T_{i-1}$ be the cluster that contains $x$ before the $i$th merge involving $x$ and let $S_i$ be the cluster that is merged with $T_{i-1}$. We prove by induction that $\text{avg}(x, T_i - x) \leq \ln H_{|T_i|-1}\text{avg}(Y, Z)$, which implies on the desired result because $T_t = X$ for some $t$. To establish the induction, we use the triangle inequality to write $\text{avg}(x, T_i - x)$ as a function of both $\text{avg}(x, T_{i-1} - x)$ and $\text{avg}(T_{i-1}, S_i)$, and also argue that $\text{avg}(T_{i-1}, S_i) \leq \text{avg}(X, Y)$.

**Lemma 3.2.** *Let* $X$, $Y$ *and* $Z$, *with* $|X| \geq 2$ *and* $Y \neq Z$, *be clusters of* $\mathcal{A}^k$. *Then, for every* $x \in X$, *we have that* $\text{avg}(x, X) \leq \text{avg}(x, X - x) \leq H_{|X|-1}\text{avg}(Y, Z)$.

The next result is a simple consequence of the previous one.

**Theorem 3.3.** *Let* $k \geq 2$ *and let* $X$, $Y$ *and* $Z$, *with* $Y \neq Z$, *be clusters of a* $k$-*clustering built by* `average-link`. *Then,* $\text{diam}(X) \leq 2H_{|X|-1}\text{avg}(Y, Z)$.

*Proof.* If $|X| = 1$ the result holds because $\text{diam}(X) = 0$. Thus, we assume that $|X| > 1$. Let $x$ and $x'$ be such that $\text{dist}(x, x') = \text{diam}(X)$. We have that

$$\text{dist}(x, x') \leq \text{avg}(x, X) + \text{avg}(X, x') \leq 2H_{|X|-1}\text{avg}(Y, Z)$$

where the first inequality follows from the triangle inequality and the second one due to Lemma 3.2. □

The next theorem shows that `cs-ratio`$_{\text{DM}}(\mathcal{A}^k) \leq 2H_n$ and that `average-link` has a logarithmic approximation for the `cs-ratio`$_{\text{DM}}$ criterion. The first upper bound is a simple consequence of Theorem 3.3. Let OPT be the minimum possible `cs-ratio`$_{\text{DM}}$. To prove the bound on the approximation we consider two cases. If OPT $\geq 1/3$ the result holds because `cs-ratio`$_{\text{DM}}(\mathcal{A}^k) \leq 2\ln n \leq 6\text{OPT}\ln n$. If OPT $< 1/3$, we argue that the clusters in the optimal clustering are "well separated" and, hence, `average-link` builds the optimal clustering.

**Theorem 3.4.** *For all* $k$, *the* $k$-*clustering* $\mathcal{A}^k$ *built by* `average-link` *satisfies* `cs-ratio`$_{\text{DM}}(\mathcal{A}^k) \leq 2H_n$. *Furthermore, for all* $k$, `cs-ratio`$_{\text{DM}}(\mathcal{A}^k)$ *is* $O(\log n) \cdot OPT$ *where OPT is* `cs-ratio`$_{\text{DM}}$ *of the* $k$-*clustering with minimum possible* `cs-ratio`$_{\text{DM}}$.

*Proof.* The inequality $\texttt{cs-ratio}_{\texttt{DM}}(\mathcal{A}^k) \leq 2H_n$ is obtained by using Theorem 3.3, with $X$ being the cluster with the largest diameter in $\mathcal{A}^k$ and $Y$ and $Z$ being the clusters in $\mathcal{A}^k$ that satisfy $\texttt{avg}(Y,Z) = \texttt{sep}_{\texttt{min}}(\mathcal{A}^k)$.

Now we prove that $\mathcal{A}^k$ has logarithmic approximation. If OPT $\geq 1/3$, then $\texttt{cs-ratio}_{\texttt{DM}}(\mathcal{A}^k) \leq 2H_n \leq 6\text{OPT}H_n$ and, hence, the desired result holds.

Thus, we assume OPT $< 1/3$, Let $\mathcal{C}^*(k)$ be a $k$-clustering that satisfies $\texttt{cs-ratio}_{\texttt{DM}}(\mathcal{C}^*(k)) = \text{OPT}$. The following claim will be useful.

*Claim* 1. Let $C, C'$ be two clusters in $\mathcal{C}^*(k)$ and let $a, b$ be two closest points in $C$ and $C'$, that is, $\texttt{dist}(a,b) = \min\{\texttt{dist}(x,y)|(x,y) \in C \times C'\}$. Thus, $\texttt{dist}(a,b) > \max\{\texttt{diam}(C), \texttt{diam}(C')\}$.

*Proof of the claim.* We assume w.l.o.g. that $\texttt{diam}(C) \geq \texttt{diam}(C')$. For the sake of reaching a contradiction, assume that $\texttt{dist}(a,b) \leq \texttt{diam}(C)$. Then, it follows from the triangle inequality that the maximum distance between a point in $C$ and $C'$ is at most $3\texttt{diam}(C)$. Thus, $\texttt{sep}_{\texttt{min}}(\mathcal{C}^*(k)) \leq \texttt{avg}(C,C') \leq 3\texttt{diam}(C)$ and so $\texttt{cs-ratio}_{\texttt{DM}}(\mathcal{C}^*(k)) \geq \texttt{diam}(C)/3\texttt{diam}(C) = 1/3$, which contradicts our assumption. $\square$

Now, we argue that $\texttt{average-link}$ constructs the clustering $\mathcal{C}^*(k)$ when $\texttt{cs-ratio}_{\texttt{DM}}(\mathcal{C}^*(k)) < 1/3$, so its approximation is 1 in this case. For the sake of reaching a contradiction, let us assume $\mathcal{A}^k \neq \mathcal{C}^*(k)$. Hence, at some iteration $\texttt{average-link}$ merges two clusters, say $A$ and $B$, that satisfy the following properties: $A \subseteq C$ and $B \subseteq C'$, where $C$ and $C'$ are two different clusters in $\mathcal{C}^*(k)$. Let $t$ be the first iteration of $\texttt{average-link}$ when it occurs.

Case 1) $A \subset C$ or $B \subset C'$. Let us assume w.l.o.g. that $A \subset C$. In this case, there is a cluster $A'$ at the beginning of iteration $t$ such that $A' \cup A \subseteq C$. We have that $\texttt{avg}(A,A') \leq \texttt{diam}(C)$ and by the above claim the minimum distance between $A$ and $B$ is larger than $\max\{\texttt{diam}(C), \texttt{diam}(C')\}$. Thus, $\texttt{avg}(A,B) > \max\{\texttt{diam}(C), \texttt{diam}(C')\} \geq \texttt{avg}(A,A')$, which contradicts the choice of $\texttt{average-link}$.

Case 2) $A = C$ and $B = C'$. If $k = 2$ we are done. Otherwise, there exists a cluster $C'' \in \mathcal{C}^*(k)$ and two clusters $X$ and $Y$ at the beginning of iteration $t$ such that $X \cup Y \subseteq C''$. Thus, it follows from the condition OPT $< 1/3$ that $\texttt{avg}(X,Y) \leq \texttt{diam}(C'') < \frac{1}{3}\texttt{sep}_{\texttt{min}}(\mathcal{C}^*(k)) \leq \frac{1}{3}\texttt{avg}(C,C') \leq \texttt{avg}(C,C')$, which again contradicts the choice of $\texttt{average-link}$. $\square$

It is noteworthy that, in contrast to Theorem 3.1, the assumption that the points lie in a metric space is necessary to prove Theorem 3.4. In Section B.4 we present an instance that supports this observation.

Now, we present an instance, denoted by $\mathcal{I}^{CS}$, that shows that the above results are nearly tight. This instance with small modifications will also be used to investigate the tightness of our results regarding the separability (Section 4) and the cohesion (Section 5) of $\texttt{average-link}$. We note that in most of the instances presented here, including $\mathcal{I}^{CS}$, will have more than one possible execution for the methods we analyze. In these cases, we will always consider the execution that is more suitable for our purposes. These multiple executions can be avoided at the price of more complicated descriptions that involve the addition of small values $\epsilon$ to the distance or points to break ties.

Let $t$ be an integer that satisfies $t! = n$; note that $t = \Omega(\frac{\log n}{\log\log n})$. Moreover, let $A_0$ be a set containing a single point located at position $p_0$ in the real line and $A_i$, for $0 < i \leq t-1$, be a set of $(i+1)! - i!$ points that are located at position $p_i$ of the real line. We define $B_0 = A_0$ and $B_i = B_{i-1} \cup A_i$, for $i \geq 1$. Set $p_0 = 0, p_1 = 1$ and, for $i > 1$, $p_i = p_{i-1} + \texttt{avg}(A_{i-1}, B_{i-2})$. The set of points for our instance $\mathcal{I}^{CS}$ is $B_{t-1}$ and the distance between a point in $A_i$ and a point in $A_j$ is $|p_i - p_j|$. The following lemma gives properties of $\mathcal{I}^{CS}$ and, in particular, how $\texttt{average-link}$ behaves on it.

**Lemma 3.5.** *For $i \geq 0$, we have that $|B_i| = (i+1)!$ and for $i \geq 2$, we have $\texttt{diam}(B_{i-2}) = i(i-1)/2$, $\texttt{avg}(B_{i-2}, A_{i-1}) = i+1$ and $p_i = i(i+1)/2$. Furthermore, for $k \leq t$, $\texttt{average-link}$ obtains the $k$-clustering $\mathcal{A}^k = (B_{t-k}, A_{t-k+1}, \ldots, A_{t-1})$ and, in particular, for $k = 2$ it obtains the clustering $\mathcal{A}^2 = (B_{t-2}, A_{t-1})$.*

From Lemma 3.5, we have that $\texttt{sep}_{\texttt{min}}(\mathcal{A}^2) = \texttt{avg}(B_{t-2}, A_{t-1}) = t+1$ and $\texttt{diam}(B_{t-2}) = t(t-1)/2$, so $\texttt{cs-ratio}_{\texttt{DM}} = \frac{t(t-1)}{2(t+1)}$, which is $\Omega(\frac{\log n}{\log\log n})$.

Furthermore, for the clustering $\mathcal{A}' = (A_0, B_{t-1} - A_0)$ we have that

$$\texttt{sep}_{\min}(\mathcal{A}') = \texttt{avg}(A_0, B_{t-1} - A_0) \geq \frac{|A_{t-1}|}{|B_{t-1}|} \texttt{avg}(A_0, A_{t-1}) = \left( \frac{t! - (t-1)!}{t!} \right) p_{t-1} = \frac{(t-1)^2}{2} \tag{5}$$

and $\texttt{max-diam}(\mathcal{A}') \leq \texttt{diam}(B_{t-1}) = (t+1)(t+2)/2$. Thus, $\texttt{cs-ratio}_{\text{DM}}(\mathcal{A}') = O(1)$ and the logarithmic approximation of $\texttt{average-link}$ to $\texttt{cs-ratio}_{\text{DM}}$ is also nearly tight.

## 4 Separability criteria

In this section, we investigate the separability of $\texttt{average-link}$. Recall that $\text{OPT}_{\text{SEP}}(k)$ is the maximum possible $\texttt{sep}_{\text{av}}$ of a $k$-clustering for $(\mathcal{X}, \texttt{dist})$. We show that for $\texttt{average-link}$ $\texttt{sep}_{\text{av}}$ is at least $\frac{\text{OPT}_{\text{SEP}}(k)}{k + 2 \ln n}$ and that this bound is nearly tight. We also show that there are instances in which the $\texttt{sep}_{\text{av}}$ of $\texttt{single-linkage}$ and $\texttt{complete-linkage}$ are exponentially smaller than that of $\texttt{average-link}$.

Theorem 4.2 gives an upper bound on $\texttt{sep}_{\text{av}}$ for $\texttt{average-link}$ and its complete proof can be found in Section D.2. Here, we give an overview of the proof for the case $k > 2$, which is the most involved one. The proof uses the fact established by Proposition 4.1 that there exists a set of $k$ points $P \subseteq \mathcal{X}$ that satisfies $\texttt{avg}(P) \geq \text{OPT}_{\text{SEP}}(k)$. This holds because a set of $k$ randomly selected points that intersect all clusters of a $k$-clustering with maximum $\texttt{sep}_{\text{av}}$ satisfies the the desired property (in expectation). Having this result in hands, it is enough to show that $\texttt{avg}(P)$ is $O((k + H_{n-1}) \texttt{sep}_{\text{av}}(\mathcal{A}^k))$.

This bound on $\texttt{avg}(P)$ is obtained by relating the distance of each pair of points $p, p' \in P$ with the average distance between clusters in $\mathcal{A}^k$. Let $p, p' \in P$ and let $A$ and $A'$ be clusters in $\mathcal{A}^k$ such that $p \in A$ and $p' \in A'$. Moreover, let $S$ be a cluster in $\mathcal{A}^k$, with $S \notin \{A, A'\}$. From the triangle inequality we have that $\texttt{dist}(p, p') = \texttt{avg}(p, p') \leq \texttt{avg}(p, A) + \texttt{avg}(A, S) + \texttt{avg}(S, A') + \texttt{avg}(A', p')$. Then, by bounding both $\texttt{avg}(p, A)$ and $\texttt{avg}(A', p')$ via Lemma 3.2, with $Y$ and $Z$ satisfying $\texttt{avg}(Y, Z) \leq \texttt{sep}_{\text{av}}(\mathcal{A}^k)$, we conclude that $\texttt{dist}(p, p') \leq 2H_n \texttt{sep}_{\text{av}}(\mathcal{A}^k) + \texttt{avg}(A, S) + \texttt{avg}(S, A')$. In general lines, the result is then established by averaging this inequality for all $S \notin \{A, A'\}$ and for all $p, p' \in P$.

**Proposition 4.1.** *There is a set of points $P \subseteq \mathcal{X}$ with the following properties: $|P| = k$ and $\texttt{avg}(P) \geq OPT_{\text{SEP}}(k)$.*

**Theorem 4.2.** *For every $k$, the $k$-clustering $\mathcal{A}^k$ obtained by $\texttt{average-link}$ satisfies $\texttt{sep}_{\text{av}}(\mathcal{A}^k) \geq \frac{OPT_{\text{SEP}}(k)}{k + 2H_n}$.*

We present two instances that, together, show that the previous theorem is nearly tight. The first is the instance $\mathcal{I}^{CS}$ presented right after Theorem 3.4. For $\mathcal{I}^{CS}$, the clustering $\mathcal{A}^2 = (A_{t-1}, B_{t-2})$ built by $\texttt{average-link}$ satisfies $\texttt{sep}_{\text{av}}(\mathcal{A}^2) = \texttt{avg}(A_{t-1}, B_{t-2}) = t + 1$. On the other hand, Eq. (5) shows that $\texttt{sep}_{\text{av}}(\mathcal{A}') = \frac{(t-1)^2}{2}$, for the clustering $\mathcal{A}' = (A_0, B_{t-1} - A_0)$. Thus, for $\mathcal{I}^{CS}$, $\texttt{sep}_{\text{av}}(\mathcal{A}^2)$ is $O(\frac{\text{OPT}_{\text{SEP}}(k) \log \log n}{\log n})$.

Now, we present our second instance, denoted by $\mathcal{I}_k^{sep}$. Let $k$ be an odd number and let $D$ and $\epsilon$ be positive numbers. The set of points of $\mathcal{I}_k^{sep}$ is given by $S_1 \cup S_2 \cup S_3$, where $|S_1| = |S_2| = (k-1)/2$ and $S_3 = \{s_i | 1 \leq i \leq k-2\}$. We have $\texttt{dist}(x, y) = \epsilon$ for $x, y \in S_1$, $\texttt{dist}(x, y) = \epsilon$ for $x, y \in S_2$, $\texttt{dist}(x, y) = 1$ for $x, y \in S_3$ and $\texttt{dist}(x, y) = D$ if $x$ and $y$ are not in the same set.

For $\mathcal{I}_k^{sep}$, when $D$ is sufficiently large and $\epsilon$ is sufficiently small, $\mathcal{A}^k = (S_1, S_2, s_1, \ldots, s_{k-2})$ and $\texttt{sep}_{\text{av}}(\mathcal{A}^k) = O(D/k)$. On the other hand, the $\texttt{sep}_{\text{av}}$ of the $k-$clustering that has the cluster $S_3$ and $k - 1$ singletons corresponding to the points in $S_1 \cup S_2$ is $\Omega(D)$. Thus, $\texttt{sep}_{\text{av}}(\mathcal{A}^k)$ is $O(\text{OPT}_{\text{SEP}}(k)/k)$.

We note that $\texttt{single-linkage}$ and $\texttt{complete-linkage}$ also obtain the $k$-clustering $\mathcal{A}^k$ for $\mathcal{I}_k^{sep}$, so the upper bound $\text{OPT}_{\text{SEP}}(k)/k$ also holds for them. In Section D.3 we present instances that show that $\texttt{sep}_{\text{av}}$ is $O(\frac{\text{OPT}_{\text{SEP}}(k)}{\sqrt{n}})$ for both $\texttt{single-linkage}$ and $\texttt{complete-linkage}$.

The instance $\mathcal{I}_k^{sep}$ is particularly interesting because it also shows that natural cohesion and separability criteria can be conflicting. The key reason is that any method $M$ with bounded approximation

(in terms of $n$) regarding max-diam or to max-avg (Equation 3) has to build the $k$-clustering $\mathcal{A}^k$ for $\mathcal{I}_k^{sep}$. Thus, by analysing $\mathcal{I}_k^{sep}$ we can conclude that the approximation factor of $M$ to $\mathtt{sep_{av}}$ is $O(1/k)$ and to $\mathtt{sep_{min}}$ is $O(1/D)$. The details can be found in Section D.4.

# 5 On the cohesion of average-link

In this section, we prove that $\mathtt{max\text{-}diam}(\mathcal{A}^k) \leq \min\{k, 1 + 4\ln n\}k^{1.59}\mathrm{OPT_{AV}}(k)$ and we also present an instance which shows that $\mathtt{max\text{-}diam}(\mathcal{A}^k) \geq k\mathrm{OPT_{DM}}(k)$.

Dasgupta and Laber [2024] presented an interesting approach to devise upper bounds on cohesion criteria for a class of linkage methods that includes average-link. Although this approach was used to show that the maximum pairwise average distance of a cluster in $\mathcal{A}^k$ is at most $k^{1.59}\mathrm{OPT_{AV}}(k)$, it cannot be employed, at least directly, to bound the maximum diameter of a cluster in $\mathcal{A}^k$. Thus, to obtain our $(1 + 4\ln n)k^{1.59}\mathrm{OPT_{AV}}(k)$ bound we combine the results of [Dasgupta and Laber, 2024] with Theorem 3.4 while for the $k^{1+1.59}\mathrm{OPT_{AV}}(k)$ bound we add some new ideas/analysis on top of those from [Dasgupta and Laber, 2024].

The analysis in Dasgupta and Laber [2024] keeps a dynamic partition of the clusters produced by the linkage method under consideration. Each group in the partition is a set of clusters denoted by *family*. A point $p$ belongs to a family $F$ if it belongs to some cluster in $F$. Thus, $\mathtt{diam}(F)$ is given by the maximum distance among the points that belong to $F$. The approach bounds the diameter of each family $F$ as (essentially) a function of the clusters that $F$ touches in a target $k$-clustering $\mathcal{T} = (T_1, \ldots, T_k)$. The bound on $\mathtt{diam}(F)$ is then used to upper bound the diameter of the clusters in $F$. For a $k$-clustering $\mathcal{C}$, let $\mathtt{avg\text{-}diam}(\mathcal{C}) := \frac{1}{k}\sum_{i=i}^{k} \mathtt{diam}(C_i)$. As in Dasgupta and Laber [2024], we use as the target clustering the one with minimum $\mathtt{avg\text{-}diam}$.

We explain how the families evolve along the execution of a linkage method, in particular average-link. Initially, we have $k$ families, $F_1, \ldots, F_k$, where $F_i$ is a family that contains $|T_i|$ clusters, each one being a point from $T_i$. Furthermore, the families are organized in a directed forest $D$ that initially consists of $k$ isolated nodes, where the $i$th node corresponds to family $F_i$.

We specify how the families and the forest $D$ are updated when the linkage method merges the clusters $g$ and $g'$ belonging to the families $F$ and $F'$, respectively. Assume w.l.o.g. $|F| \geq |F'|$. We have the following cases:

case 1 $|F'| = 1$ and $|F| > 1$. In this case two new families are created, $F^{new} := F - \{g\}$ and $F^{new'} := \{g \cup g'\}$. Moreover, $F^{new}$ and $F^{new'}$ become, respectively, parents of $F$ and $F'$ in $D$

case 2 $|F'| > 1$ or $|F| = 1$. In this case, only one family is created, $F^{new} := (F \cup F' \cup \{g \cup g'\}) - g - g'$. Moreover, $F^{new}$ becomes parent of both $F$ and $F'$ in $D$.

We say that a family $F$ is *regular* if $|F| > 1$.

**Proposition 5.1** (Proposition 3.1 of Dasgupta and Laber [2024]). *At the beginning of each iteration of* average-link *at least one of the roots of the forest $D$ corresponds to a regular family.*

Let $\mathcal{M}$ be the class of linkage methods (Algorithm 2) whose function $f$, employed to measure the distance between clusters $A$ and $B$ satisfies

$$\{\mathtt{dist}(a,b)|(a,b) \in A \times B\} \leq f(A,B) \leq \mathtt{diam}(A \cup B) \tag{6}$$

**Proposition 5.2** (Proposition 5.1 of Dasgupta and Laber [2024]). *The diameter of every regular family $F$ produced along the execution of a linkage method in $\mathcal{M}$ is at most $k^{\log_2 3}OPT_{AV}(k)$.*

Note that the function $\mathtt{dist}_{AL}$ employed by average-link satisfies the condition given by (6) and, thus, the above proposition holds for average-link.

We are ready to establish the main result of this section.

**Theorem 5.3.** *Every cluster $S$ in $\mathcal{A}^k$ satisfies* $\mathtt{diam}(S) \leq \min\{k, 4\ln n + 1\}k^{\log_2 3}OPT_{AV}(k)$.

*Proof.* Let $V = \{T \in \mathcal{T}|S \cap T \neq \emptyset\}$ be the set of clusters of the target clustering $\mathcal{T}$ that intersect $S$. We build a graph $G$ whose nodes correspond to the clusters in $V$. At the beginning of average-link's execution, $G$ contains the set of nodes $V$ and no edges.

At each iteration, there are two possibilities for the clusters $g$ and $g'$ that are merged by `average-link`: $(g \cup g') \cap S = \emptyset$ or $(g \cup g') \subseteq S$. We define how $G$ is updated in each case:

Case 1) $(g \cup g') \cap S = \emptyset$. In this case, $G$ is not updated.

Case 2) $(g \cup g') \subseteq S$. Let $x$ and $y$ be points in $g$ and $g'$ such that $\mathtt{dist}(x,y)$ is minimum and let $T^x$ and $T^y$ be the clusters in $\mathcal{T}$ that contain $x$ and $y$, respectively. We add an edge of weight $\mathtt{dist}(x,y)$ between $T^x$ and $T^y$. We say, in this case, that $x$ and $y$ are *associated* with the edge that links $T^x$ to $T^y$.

We need the following two claims:

*Claim* 2. $\mathtt{dist}(x,y) \leq k^{\log_2 3} \mathrm{OPT}_{\mathtt{AV}}(k)$.

*Proof of the claim.* Let $H$ be a regular family at the beginning of iteration $t$ Such family does exist due to Proposition 5.1. Moreover, let $h$ and $h'$ be two clusters in $H$. We have that

$$\mathtt{dist}(x,y) \leq \mathtt{dist}_{AL}(g,g') \leq \mathtt{dist}_{AL}(h,h') \leq \mathtt{diam}(h \cup h') \leq \mathtt{diam}(H) \leq k^{\log_2 3} \mathrm{OPT}_{\mathtt{AV}}(k),$$

where the second inequality holds by the choice of `average-link` and the last inequality holds due to the Proposition 5.2. $\square$

*Claim* 3. For a cluster $C$, let $V_C := \{T \in \mathcal{T} | T \cap C \neq \emptyset\}$. Let $S'$ be a cluster generated by `average-link` that is a subset of $S$. Then, when $S'$ is created, the subgraph of $G$ induced by $V_{S'}$ is connected.

*Proof of the claim* If $|S'| = 1$ the property holds. Let $S'$ be a cluster obtained by merging $S_1$ and $S_2$. By induction, the property holds for $S_1$ and $S_2$. Since an edge is added between nodes in $V_{S_1}$ and $V_{S_2}$ then the property also holds for $S$. $\square$

Thus, at the end of the algorithm, $G$ is connected and each of its edges has weight at most $k^{\log_2 3} \mathrm{OPT}_{\mathtt{AV}}(k)$. Let $x$ and $y$ be points in $S$ such that $\mathtt{dist}(x,y) = \mathtt{diam}(S)$ and let $T^x = v_1 \ldots v_\ell = T^y$ be a path in $G$ from $T^x$ to $T^y$.

Consider a sequence of points $x = p_1 p_1' \ldots p_\ell p_\ell' = y$, where $p_i$ and $p_i'$ are the points in $v_i$ associated with the edge $v_{i-1} v_i$ and $v_i v_{i+1}$, respectively. From the triangle inequality

$$\mathtt{dist}(x,y) \leq \sum_{i=1}^{\ell-1} \mathtt{dist}(p_i', p_{i+1}) + \sum_{i=1}^{\ell} \mathtt{dist}(p_i, p_i') \leq (k-1) k^{\log_2 3} \mathrm{OPT}_{\mathtt{AV}}(k) + \sum_{i=1}^{k} \mathtt{diam}(T_i) \leq$$

$$(k-1) k^{\log_2 3} \mathrm{OPT}_{\mathtt{AV}}(k) + k \mathrm{OPT}_{\mathtt{AV}}(k)$$

For the logarithmic bound, let $S_1$ and $S_2$ be the two clusters that are merged to form $S$. At the beginning of the iteration in which $S_1$ and $S_2$ are merged, Proposition 5.1 assures that there exists a regular family, say $H$. Let $h$ and $h'$ be two clusters in $H$. By Proposition 5.2, $\mathtt{avg}(h,h') \leq \mathtt{diam}(H) \leq k^{\log_2 3} \mathrm{OPT}_{\mathtt{AV}}(k)$. Thus, by Theorem 3.3, $\mathtt{diam}(S_1) \leq 2 \ln n \cdot \mathtt{avg}(h,h') \leq 2 \ln n \cdot k^{\log_2 3} \mathrm{OPT}_{\mathtt{AV}}(k)$ and $\mathtt{diam}(S_2) \leq 2 \ln n \cdot k^{\log_2 3} \mathrm{OPT}_{\mathtt{AV}}(k)$. Let $s_1 \in S_1$ and $s_2 \in S_2$ be such that $\mathtt{dist}(s_1, s_2) = \min\{\mathtt{dist}(p,q) | (p,q) \in S_1 \times S_2\}$. Since $S_1$ and $S_2$ are merged we have that $\mathtt{dist}(s_1, s_2) \leq \mathtt{avg}(S_1, S_2) \leq \mathtt{avg}(h,h') \leq k^{\log_2 3} \mathrm{OPT}_{\mathtt{AV}}(k)$. Thus, $\mathtt{diam}(S) \leq \mathtt{diam}(S_1) + \mathtt{dist}(s_1, s_2) + \mathtt{diam}(S_1) \leq (1 + 4 \ln n) k^{\log_2 3} \mathrm{OPT}_{\mathtt{AV}}(k)$. $\square$

Theorem 3.4 of Dasgupta and Laber [2024] presents an instance with $n = 2k - 2$ points for which `single-linkage` builds a $k$-clustering that has a cluster whose diameter is $\Omega(k^2 \mathrm{OPT}_{\mathtt{AV}}(k))$. Thus, this result together with Theorem 5.3 show a separation between `average-link` and `single-linkage` when $k$ is $\Omega(\log^{2.41} n)$.

Our last theoretical result is a lower bound on the maximum diameter of the clustering built by `average-link`. Its proof can be found in the Section E and it employs an augmented version of instance $\mathcal{I}^{CS}$, presented right after Theorem 3.4.

**Theorem 5.4.** *There is an instance for which the $k$-clustering $\mathcal{A}^k$ built by* `average-link` *satisfies* $\mathtt{max\text{-}diam}(\mathcal{A}^k) \in \Omega(k \mathrm{OPT}_{\mathtt{DM}}(k))$

Table 1: Average ratio between the result of a method and the best one for each criterion and each group of $k$. The best results are bold-faced

| | Small | | | Medium | | | Large | | |
|---|---|---|---|---|---|---|---|---|---|
| | A | C | S | A | C | S | A | C | S |
| sep$_{\text{min}}$ | **0,99** | 0,82 | 0,76 | **1** | 0,81 | 0,68 | **1** | 0,81 | 0,72 |
| sep$_{\text{av}}$ | **0,97** | 0,82 | 0,94 | 0,97 | 0,9 | **1** | 0,98 | 0,96 | **1** |
| max-diam | 0,85 | **1** | 0,72 | 0,8 | **1** | 0,48 | 0,76 | **1** | 0,38 |
| max-avg | 0,95 | **0,96** | 0,86 | **0,99** | 0,89 | 0,71 | **0,99** | 0,84 | 0,67 |
| cs-ratio$_{\text{DM}}$ | **0,96** | 0,92 | 0,63 | 0,95 | **0,97** | 0,4 | 0,93 | **0,99** | 0,33 |
| cs-ratio$_{\text{AV}}$ | **0,98** | 0,82 | 0,69 | **1** | 0,73 | 0,51 | **1** | 0,68 | 0,4 |

# 6 Experiments

In this final section, we briefly present an experiment in which we evaluate whether `average-link`, in addition to having better theoretical bounds, it also has a better performance in practice for the studied criteria. We employed 10 datasets and used the Euclidean metric to measure distances. For each of them, we executed `average-link`, `complete-linkage` and `single-linkage`, for the following sets of values of $k$: $\text{Small}=\{k|2 \leq k \leq 10\}$, $\text{Medium}=\{k|\sqrt{n}-4 \leq k \leq \sqrt{n}+4\}$ and $\text{Large}=\{k|k = n/i \text{ and } 2 \leq i \leq 10\}$. More details, as well as the results of our experiment with other distances, can be found in Section F.

Table 6 shows the average ratio between the result of a method and that of the best one, grouped by criterion and set of $k$. Each entry is the average of 90 ratios (9 $k'$s and 10 datasets) and each of these ratios for a method $M$ is a value between 0 and 1 that is obtained by dividing the minimum between the result of $M$ and that of the best method by the maximum between them. The letters A, C and S are the initials of the evaluated methods.

Concerning separability criteria, `single-linkage` and `average-link` have the best results for sep$_{\text{av}}$. The latter has some advantage when $k$ is small, which is in line with its better worst-case bound for small $k$ (results from Section 4). For sep$_{\text{min}}$, `average-link` has a huge advantage, which is not surprising since its linkage rule tries to increase sep$_{\text{min}}$ at each step by merging the the clusters $A$ and $B$ for which $\text{avg}(A, B) = \text{sep}_{\text{min}}(\mathcal{C})$, where $\mathcal{C}$ is the current clustering.

Regarding cohesion criteria, `complete-linkage` and `average-link` were the best methods. They had close results for max-avg while for max-diam the former had a strong dominance. These results align with ours and those from [Dasgupta and Laber, 2024], in the sense that they show that these linkage methods present better worst-case upper bounds than `single-linkage` when the comparison is made against $\text{OPT}_{\text{AV}}(k)$. Moreover, the advantage of `complete-linkage` for max-diam is also expected since it is the "natural" greedy rule to minimize the maximum diameter (See Proposition 2.1 of Dasgupta and Laber [2024]).

For cs-ratio$_{\text{DM}}$, `average-link` and `complete-linkage` present the best results, with the former being slightly superior for the small $k$ and the latter being slightly superior when $k$ is not small. `average-link` has a huge dominance for the cs-ratio$_{\text{AV}}$ criterion, which lines up with the theoretical results from Section 3.1.

In summary, these experiments, together with our theoretical results, provide evidence that `average-link` is a better choice when both cohesion and separability are relevant.

**Acknowledgements** The work of the first author is partially supported by CNPq (grant 310741/2021-1). This study was financed in part by the Coordenação de Aperfeiçoamento de Pessoal de Nível Superior - Brasil (CAPES) - Finance Code 001

**Limitations.** We have not identified a major limitation in our work. That said, the assumption that the points lie in a metric space used in our results (except Theorem 3.1) could be seen as a limitation. On the experimental side, having more than 10 datasets would give our conclusions more robustness.

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

## A  Proof of proposition 2.1

*Proof.* Let $a \in A$ and $c \in C$. Then, $\texttt{dist}(a, c) \leq \texttt{dist}(a, b) + \texttt{dist}(b, c)$ for every $b \in B$. Thus,

$$|B|\texttt{dist}(a,c) \leq \sum_{b \in B}(\texttt{dist}(a,b) + \texttt{dist}(b,c))$$

It follows that

$$|B| \sum_{a \in A} \sum_{c \in C} \texttt{dist}(a,c) \leq \sum_{a \in A} \sum_{c \in C} (\sum_{b \in B}(\texttt{dist}(a,b) + \texttt{dist}(b,c))) =$$

$$|C| \sum_{a \in A} \sum_{b \in B} \texttt{dist}(a,b) + |A| \sum_{b \in B} \sum_{c \in C} \texttt{dist}(b,c)$$

Dividing both sides by $|A| \cdot |B| \cdot |C|$ we establish the inequality. $\qquad\square$

## B  Proofs of section 3

### B.1  Proof of Theorem 3.1

*Proof.* When $k = n$ the result is valid because $\texttt{avg}(A^n) = 0$ for every $A \in \mathcal{A}^n$. We assume by induction that the result holds for $k + 1$ and we prove that it also holds for $k$. Let $A$ and $B$ be the clusters in $\mathcal{A}^{k+1}$ that are merged to obtain $\mathcal{A}^k$, so $\mathcal{A}^k = \mathcal{A}^{k+1} \cup (A \cup B) - \{A, B\}$. Let $S, T$ and $U$ be clusters in $\mathcal{A}^k$, with $T \neq U$. It is enough to prove that $\texttt{avg}(S) \leq \texttt{avg}(T, U)$.

Case 1) $A \cup B \notin \{S, T, U\}$. In this case, $S, T, U \in \mathcal{A}^{k+1}$ and, then, by induction, $\texttt{avg}(S) \leq \texttt{avg}(T, U)$.

Case 2) $A \cup B = S$ and $S \notin \{T, U\}$. Since $A, B, T, U \in \mathcal{A}^{k+1}$, the induction hypothesis assures that $\texttt{avg}(A) \leq \texttt{avg}(T, U)$ and $\texttt{avg}(B) \leq \texttt{avg}(T, U)$ and the $\texttt{average-link}$ rule ensures that $\texttt{avg}(A, B) \leq \texttt{avg}(T, U)$. Since $\texttt{avg}(S)$ is a convex combination of $\texttt{avg}(A), \texttt{avg}(B)$ and $\texttt{avg}(A, B)$, the above inequalities imply that $\texttt{avg}(S) = \texttt{avg}(A \cup B) \leq \texttt{avg}(T, U)$.

Case 3) $A \cup B = S$ and $S \in \{T, U\}$. We assume w.l.o.g. that $S = T$. The induction hypothesis and the $\texttt{average-link}$ rule guarantee that $\max\{\texttt{avg}(A), \texttt{avg}(B), \texttt{avg}(A, B)\} \leq \min\{\texttt{avg}(A, U), \texttt{avg}(B, U)\}$ Since $\texttt{avg}(S, U)$ is a convex combination of $\texttt{avg}(A, U)$ and $\texttt{avg}(B, U)$ and $\texttt{avg}(S)$ is a convex combination of $\texttt{avg}(A), \texttt{avg}(B)$ and $\texttt{avg}(A, B)$, the above inequality implies that $\texttt{avg}(S) = \texttt{avg}(A \cup B) \leq \texttt{avg}(T, U)$.

Case 4) $S \neq A \cup B$ and $A \cup B \in \{T, U\}$. We assume w.l.og. that $T = A \cup B$. Since $S, A, B, U \in \mathcal{C}^{k+1}$, the induction hypothesis assures that $\texttt{avg}(S) \leq \min\{\texttt{avg}(A, U), \texttt{avg}(B, U)\}$ Since $\texttt{avg}(T, U)$ is a convex combination of $\texttt{avg}(A, U)$ and $\texttt{avg}(B, U)$, the above inequality assures that $\texttt{avg}(S) \leq \texttt{avg}(T, U)$. $\qquad\square$

### B.2  Lower bounds on $\texttt{cs-ratio}_{\text{AV}}$ for other methods

The following examples show that the $\texttt{cs-ratio}_{\text{AV}}$ of $\texttt{complete-linkage}$, $\texttt{single-linkage}$ and a random hierarchy can be much higher than that of $\texttt{average-link}$ in metric spaces.

**single-linkage.** Consider the instance with $n$ points $x_1, \ldots, x_n$ in the real line, where $x_i = 1$, if $i = 1$, and $x_i = x_{i-1} + 1 - i\epsilon$, for $i > 1$. For $\epsilon$ sufficiently small, $\texttt{single-linkage}$ builds the $k$-clustering $\mathcal{C} = (x_1, x_2, \ldots, x_{k-1}, \{x_k, \ldots, x_n\})$. We have that $\texttt{avg}(\{x_k, \ldots, x_n\})$ is $\Omega(n - k)$ while $\texttt{avg}(x_1, x_2) = 1 - \epsilon$, so that $\texttt{cs-ratio}_{\text{AV}}(\mathcal{C})$ is $\Omega(n - k)$.

**complete-linkage.** Let $t = 2^m - 1$, where $m$ is a positive integer and let $p = 2(t^2 + t)$. We build an instance whose set of points $\mathcal{X} = A \cup B \cup C \cup D \cup E$ has $n = 2p$ points, where $A, B, C, D$ and $E$ are sets of points in $\mathbb{R}^{p+1}$ that satisfy the following properties:

- the first coordinate of the points in $A \cup B \cup C \cup D$ is the only one that has a value different than 0;
- $A = \{a_1, \ldots, a_t\}$ and the first coordinate of $a_i$ is equal to $i + 1/2$;

- $B = \{b_1, \ldots, b_t\}$ and the first coordinate of $b_i$ is equal to $-(i + 1/2)$;
- $C$ has $t^2$ points and all have the first coordinate $1/2$;
- $D$ has $t^2$ points and all have the first coordinate $-1/2$;
- $E = \{e_1, \ldots, e_p\}$, where the value of the first coordinate of $e_i$ is $t^2$, the $(i+1)$th coordinate has value $1.5t$ and all other coordinates have value equal to $0$.

The distance between any two points in $\mathcal{X}$ is given by the $\ell_1$ metric. Hence, the distance between any two points in $E$ is $3t$, the distance between points in $A \cup B \cup C \cup D$ is at most $2t + 1$ and the distance between a point in $A \cup B \cup C \cup D$ and a point in $E$ is at least $t^2$. For $i \leq p$, let $E_i = \{e_i, \ldots, e_p\}$.

Thus, for $2 < k < p = n/2$, there is a way to break ties for which the $k$-clustering obtained by `complete-linkage` is $\mathcal{C}^k = (A \cup C, B \cup D, e_1, e_2, \ldots, e_{k-3}, E_{k-2})$.

We have that $\max\{\texttt{dist}(a, d) \in A \times D\} \leq t + 1$, $\max\{\texttt{dist}(b, c) \in B \times C\} \leq t + 1$ and $\max\{\texttt{dist}(a, b) \in A \times B\} \leq 2t + 1$. Thus, we get that

$$\texttt{sep}_{\min}(\mathcal{C}^k) \leq \texttt{avg}((A \cup C, B \cup D)) \leq$$

$$\frac{1}{(t^2 + t)^2} \left( \sum_{x \in A} \sum_{y \in B} \texttt{dist}(x, y) + \sum_{x \in A} \sum_{y \in D} \texttt{dist}(x, y) + \sum_{x \in C} \sum_{y \in B} \texttt{dist}(x, y) + \sum_{x \in C} \sum_{y \in D} \texttt{dist}(x, y) \right)$$

$$\leq \frac{t^2(2t + 1) + t^3(t + 1) + t^3(t + 1) + t^4}{(t^2 + t)^2} \leq 3$$

Since $\texttt{max-avg}(\mathcal{C}) \geq \texttt{avg}(E_{k-2}) = 3t$, we get that $\texttt{cs-ratio}_{\texttt{AV}}(\mathcal{C}^k)$ is $\Omega(t)$ and, hence, $\Omega(\sqrt{n})$.

**random hierarchy.** To analyze a random hierarchy, we first need to define how it is generated. We start with a random permutation of the points in $\mathcal{X}$ and a clustering $\mathcal{C}$ containing initially the cluster comprised by all points in $\mathcal{X}$. Let $x_1, \ldots, x_n$ be the points in $\mathcal{X}$ according to the order given by the permutation. Then, we perform the following steps until we have $n$ clusters:

- $j \leftarrow$ a randomly selected a number in the set $\{1, 2, \ldots, n - 1\}$.
- If the points $x_j$ and $x_{j+1}$ are in the same cluster $C \in \mathcal{C}$
  - split $C$ into $C_{\leq} = \{x_i \in C | i \leq j\}$ and the cluster $C_{>} = C - C_{\leq}$.
  - Update $\mathcal{C}$ by replacing $C$ with $C_{\leq}$ and $C_{>}$

After $t$ splits we have a clustering with $n - t$ clusters.

Now, we consider an instance with $n$ points and 3 groups $X$, $Y$ and $Z$, that satisfy $|X| = |Y| = (n-1)/2$ and $Z = \{z\}$. The distance between any two points in $X$ is 1 and the same holds for $Y$. Moreover, the distance between points in $X$ and $Y$ is 2. The distance of $z$ to any other point is $D >> n^2$. Any $k$-clustering, with $k \geq 3$, has $\texttt{sep}_{\min} \leq 2$ because at least two clusters do not contain $z$. Let $k \leq n/2$. The probability that $z$ is a singleton in the $k$-clustering when $z \notin \{x_1, x_n\}$ is

$$\frac{\binom{n-3}{k-3}}{\binom{n-1}{k-1}} = \frac{(k-1)(k-2)}{(n-1)(n-2)} < \frac{1}{4}$$

Then, with probability at least $3/4$, there will be a cluster $C$ that contains $z$ and a point in $X \cup Y$, which implies that $E[\texttt{avg}(C)] \geq D/4n^2$. Thus, with probability at least $3/4$ the $k$-clustering induced by the random hierarchy has $\texttt{sep}_{\texttt{av}}\ \Omega(D/4n^2)$, when $z \notin \{x_1, x_n\}$. Since the probability of $z \notin \{x_1, x_n\}$ is $(n-2)/n$, the same bound holds when we drop this constraint.

## B.3 On the approximation of `average-link` for `cs-ratio`$_{\texttt{AV}}$

Let $n$ be an even number, $k = 2$ and $\epsilon$ a positive number very close to 0. Consider 4 set of points $S_1, S_2, S_3$ and $S_4$, where $S_1 = \{s_1\}$, $S_2 = \{s_2\}$ and $S_3$ and $S_4$ have $n/2 - 1$ points each. We have $\texttt{dist}(x, y) = \epsilon$ for $x, y \in S_3$, $\texttt{dist}(x, y) = \epsilon$ for $x, y \in S_4$, $\texttt{dist}(s_1, s_2) = T$ and $\texttt{dist}(x, y) = T$ for $(x, y) \in S_3 \times S_4$. In addition, we have $\texttt{dist}(s_1, x) = 2T$ for $x \neq s_2$ and $\texttt{dist}(s_2, y) = 2T$ for $y \neq s_1$.

Clearly, the 4-clustering obtained by `average-link` is $(S_1, S_2, S_3, S_4)$. Then, to obtain a 2-clustering, it merges the clusters $S_1$ and $S_2$ and, next, $S_3$ and $S_4$, so that the final 2-clustering is $\mathcal{A}^2 = (S_1 \cup S_2, S_3 \cup S_4)$, which satisfies $\texttt{max-avg}(\mathcal{A}^2) = T$ and $\texttt{sep}_{\min}(\mathcal{A}) = 2T$. On the other hand, for the clustering $\mathcal{S} = (S_1 \cup S_3, S_2 \cup S_4)$, we have that $\texttt{max-avg}(\mathcal{S})$ is $O(T/n^2)$ and $\texttt{sep}_{\min}(\mathcal{S}) \geq T$. Thus, the approximation of `average-link` is $\Omega(n^2)$

## B.4 Triangle inequality is necessary for Theorem 3.4

We present an instance that shows that the assumption that points lie in a metric space is necessary to establish Theorem 3.4.

Let $A$ and $B$ be sets with $n/2 - 1$ and $n/2$ points, respectively. We have $\texttt{dist}(a, a') = 1$ if $a, a' \in A$; $\texttt{dist}(b, b') = 1$ if $b, b' \in B$ and $\texttt{dist}(a, b) = 4$ if $(a, b) \in A \times B$. Moreover, let $p$ be a point that is not in $A \cup B$. There is a point $a \in A$ for which $\texttt{dist}(a, p) = n/2 - 2$ and for all other points $a' \in A - \{a\}$, $\texttt{dist}(a', p) = 2$. Moreover, $\texttt{dist}(p, b) = 4$ for $b \in B$.

For this instance `average-link` builds the 2-clustering $\mathcal{A}^2 = (A \cup \{p\}, B)$. We have that $\texttt{diam}(A \cup p) = n/2 - 2$ and $\texttt{avg}(A \cup p, B) = 4$, Thus, $\texttt{cs-ratio}_{\text{DM}}(\mathcal{A}^2)$ is $\Omega(n)$. On the other hand, for the clustering $\mathcal{A}' = (A, B \cup p)$, $\texttt{cs-ratio}_{\text{DM}}(\mathcal{A}')$ is $O(1)$, so the approximation of `average-link` is $\Omega(n)$.

## B.5 Proof of Lemma 3.2

*Proof.* The first inequality holds because $\texttt{avg}(x, X) = \frac{|X|-1}{|X|}\texttt{avg}(x, X - x)$. Thus, we just need to prove the second one.

Let $S_1$ be the first cluster merged with $x$ by `average-link` and let $S_i$, for $i > 1$, be the cluster merged with $S_1 \cup \cdots \cup S_{i-1}$ by `average-link`. Define $T_0 := \{x\}$ and, for $i \geq 1$, $T_i := T_{i-1} \cup S_i$.

Furthermore, define $e_i$ and $m_i$ as $e_i := \texttt{avg}(T_{i-1}, S_i)$ and $m_i := \texttt{avg}(x, T_i - x)$, respectively. Note that there is $t$ for which $T_t = X$ and, hence, $m_t = \texttt{avg}(x, X - x)$.

We have that

$$m_{i+1} = \frac{|T_i| - 1}{|T_{i+1}| - 1}\texttt{avg}(x, T_i - x) + \frac{|S_{i+1}|}{|T_{i+1}| - 1}\texttt{avg}(x, S_{i+1}) \leq \tag{7}$$

$$\frac{|T_i| - 1}{|T_{i+1}| - 1}m_i + \frac{|S_{i+1}|}{|T_{i+1}| - 1}(m_i + e_{i+1}) = m_i + \frac{|S_{i+1}|}{|T_{i+1}| - 1}e_{i+1}, \tag{8}$$

where the inequality follows from the triangle inequality.

Let us consider the beginning of the iteration in which $T_{i-1}$ and $S_i$ are merged. At this point we have $\ell \geq 1$ clusters $Y_1, \ldots, Y_\ell$ such that $Y = Y_1 \cup \cdots \cup Y_\ell$ and $\ell'$ clusters $Z_1, \ldots, Z_{\ell'}$ such that $Z = Z_1 \cup \cdots \cup Z_\ell$. Note that there exist $i$ and $j$ such that $\texttt{avg}(Y_i, Z_j) \leq \texttt{avg}(Y, Z)$. Thus, we must have $e_i \leq \texttt{avg}(Y, Z)$, otherwise `average-link` would merge $Y_i$ and $Z_j$ rather than $T_{i-1}$ and $S_i$.

To establish the result, we show by induction that $m_i \leq \texttt{avg}(Y, Z) \cdot H_{|T_i|-1}$, for $i \geq 1$. The lemma is then established by taking $i = t$, where $t$ satisfies $T_t = X$.

For $i = 1$, we have $m_1 = e_1 \leq \texttt{avg}(Y, Z) < \texttt{avg}(Y, Z) \cdot H_{|T_1|-1}$. We assume by induction that $m_{i-1} \leq \texttt{avg}(Y, Z) \cdot H_{|T_{i-1}|-1}$. By inequality (7)-(8),

$$m_i \leq m_{i-1} + e_i\frac{|S_i|}{|T_i| - 1} \leq \texttt{avg}(Y, Z)\left(\sum_{h=1}^{|T_{i-1}|-1}\frac{1}{h}\right) + \texttt{avg}(Y, Z)\left(\sum_{h=|T_{i-1}|}^{|T_i|-1}\frac{1}{h}\right) = \texttt{avg}(Y, Z) \cdot H_{|T_i|-1}$$

$\square$

## C    Proof of Lemma 3.5

*Proof.* First, we note that

$$|B_{i-1}| = \sum_{h=0}^{i-1} |A_i| = \sum_{h=0}^{i-1} (h+1)! - h! = i!,$$

for $i \geq 1$.

Moreover, for $i \geq 2$, we have that

$$\mathtt{avg}(A_i, B_{i-1}) = \frac{|A_{i-1}|}{|B_{i-1}|} \mathtt{avg}(A_i, A_{i-1}) + \frac{|B_{i-2}|}{|B_{i-1}|} \mathtt{avg}(A_i, B_{i-2}) = \tag{9}$$

$$\frac{|A_{i-1}|}{|B_{i-1}|} \mathtt{avg}(A_i, A_{i-1}) + \frac{|B_{i-2}|}{|B_{i-1}|} (\mathtt{avg}(A_i, A_{i-1}) + \mathtt{avg}(A_{i-1}, B_{i-2})) = \tag{10}$$

$$\mathtt{avg}(A_i, A_{i-1}) + \frac{|B_{i-2}|}{|B_{i-1}|} \mathtt{avg}(A_{i-1}, B_{i-2}) = \tag{11}$$

$$\left( 1 + \frac{|B_{i-2}|}{|B_{i-1}|} \right) \mathtt{avg}(A_{i-1}, B_{i-2}), \tag{12}$$

where the last identity follows because $\mathtt{avg}(A_i, A_{i-1}) = p_i - p_{i-1} = \mathtt{avg}(A_{i-1}, B_{i-2})$.

By applying the above equation successively, we conclude that

$$\mathtt{avg}(A_i, B_{i-1}) = (i+1) \cdot \mathtt{avg}(A_1, B_0) = (i+1)$$

and, hence,

$$p_i = 1 + \sum_{h=1}^{i-1} (h+1) = \frac{i(i+1)}{2}.$$

Thus,

$$\mathtt{diam}(B_{i-1}) = p_{i-1} - p_0 = p_{i-1} = \frac{i(i-1)}{2}$$

Now we show that at the beginning of the step $(n-t)+i$ `average-link` keeps a clustering that contains the cluster $B_{i-1}$ and the clusters $A_j$, for $i \leq j \leq t-1$. First, we observe that after $n-t$ steps `average-link` produces a $t$-clustering $(A_0, \ldots, A_{t-1})$ since points in the same group $A_i$ are located at the same position. We analyze what happens in the remaining $t-1$ steps.

For $i = 1$ the result holds because $B_0 = A_0$. We assume as an induction hypothesis that at beginning of the step $(n-t)+i$, we have the clusters $B_{i-1}$ and $A_j$, for $j \geq i$. By construction, for $i \leq r < s$,

$$\mathtt{avg}(A_s, A_r) = p_s - p_r > p_{i+1} - p_i = \mathtt{avg}(A_i, B_{i-1}),$$

Moreover,

$$i - 1 = \mathtt{avg}(A_i, B_{i-1}) < \mathtt{avg}(A_j, B_{i-1}),$$

for $j > i$. Thus, $\mathtt{average-link}$  prefers merging $A_i$ and $B_{i-1}$ rather than any other pair of clusters, which completes the inductive step. $\qquad \square$

## D    Proofs from section 4

### D.1    Proof of Proposition 4.1

*Proof.* Let $\mathcal{C}^* = (C_1^*, \ldots, C_k^*)$ be a $k$-clustering that maximizes $\mathtt{sep}_{\mathtt{av}}$. Let $\mathcal{Q}$ be the family of sets of points $Q$ such that $|Q| = k$ and $Q$ intersects all clusters $C_1^*, \ldots, C_k^*$. Let $P = \{p_1, \ldots, p_k\}$ be a set in $\mathcal{Q}$ that satisfies $\mathtt{avg}(P) \geq \mathtt{avg}(Q)$, for every $Q \in \mathcal{Q}$. Moreover, let $U = \{u_1, \ldots, u_k\}$ be a

set of $k$ points where $u_i$ is randomly selected from $C_i^*$. It follows from the choice of $P$ that

$$\frac{k(k-1)}{2}\mathtt{avg}(P) \geq \frac{k(k-1)}{2}E[\mathtt{avg}(U)] =$$

$$E\left[\sum_{i=1}^{k-1}\sum_{j=i+1}^{k}\mathtt{dist}(u_i, u_j)\right] = \sum_{i=1}^{k-1}\sum_{j=i+1}^{k}E\left[\mathtt{dist}(u_i, u_j)\right] = \sum_{i=1}^{k-1}\sum_{j=i+1}^{k}\mathtt{avg}(C_i^*, C_j^*) \geq$$

$$\frac{k(k-1)}{2}\mathtt{sep_{av}}(C^*)$$

$\square$

### D.2 Proof of Theorem 4.2

*Proof.* Let $P = \{p_i | 1 \leq i \leq k\}$ be the $k$ points given by Proposition 4.1 and let $h$ be a function that maps each point $p \in P$ into its cluster in $\mathcal{A}^k$. Moreover, let $Y$ and $Z$ be clusters in $\mathcal{A}^k$ that satisfy $\mathtt{avg}(Y, Z) = \mathtt{sep_{min}}(\mathcal{A}^k)$.

Let $p$ and $p'$ be distinct points in $P$. We consider two cases:

Case 1) $p$ and $p'$ belong to the same cluster $A$ in $\mathcal{A}^k$. From Theorem 3.3 we have that

$$\mathtt{dist}(p, p') \leq \mathtt{diam}(A) \leq 2H_{|A|}\mathtt{avg}(Y, Z) = 2H_{|A|}\mathtt{sep_{min}}(\mathcal{A}^k)$$

Thus,

$$\sum_{p,p'\in P\cap A}\mathtt{dist}(p, p') \leq \sum_{p,p'\in P\cap A}2H_{|A|}\mathtt{sep_{min}}(\mathcal{A}^k). \tag{13}$$

By considering all clusters $A \in \mathcal{A}^k$ we get

$$\sum_{\substack{p,p'\in P \\ h(p)=h(p')}}\mathtt{dist}(p, p') \leq \sum_{\substack{p,p'\in P \\ h(p)=h(p')}}2H_n\mathtt{sep_{min}}(\mathcal{A}^k) \tag{14}$$

Case 2) $p$ and $p'$ belong, respectively, to different clusters $A$ and $A'$ in $\mathcal{A}^k$. We consider two subcases:

*subcase 2.1)* $k = 2$. In this case, from the triangle inequality, we have that $\mathtt{dist}(p, p') = \mathtt{avg}(p, p') \leq \mathtt{avg}(p, A) + \mathtt{avg}(A, A') + \mathtt{avg}(A', p')$. By using Lemma 3.2, we have that $\mathtt{avg}(p, A) \leq H_{n-1}\mathtt{avg}(A, A') = H_{n-1}\mathtt{sep_{min}}(\mathcal{A}^k)$ and $\mathtt{avg}(p', A') \leq H_{n-1}\mathtt{avg}(A, A') = H_{n-1}\mathtt{sep_{min}}(\mathcal{A}^k)$. Thus,

$$\sum_{\substack{p,p'\in P \\ h(p)\neq h(p')}}\mathtt{dist}(p, p') = \mathtt{dist}(p, p') \leq 2H_{n-1}\mathtt{sep_{min}}(\mathcal{A}^k) + \mathtt{avg}(A, A'), \tag{15}$$

where the first identity holds because $P = \{p, p'\}$.

*subcase 2.2)* $k > 2$. Let $S$ be a cluster in $\mathcal{A}^k - \{A, A'\}$. From the triangle inequality, we have that

$$\mathtt{dist}(p, p') = \mathtt{avg}(p, p') \leq \mathtt{avg}(p, A) + \mathtt{avg}(A, S) + \mathtt{avg}(S, A') + \mathtt{avg}(A', p')$$

If $|A| = 1$, $\mathtt{avg}(p, A) = 0 \leq H_{|A|} \cdot \mathtt{sep_{min}}(\mathcal{A}^k)$. Moreover, if $|A| \geq 2$, it follows from Lemma 3.2 that $\mathtt{avg}(p, A) \leq H_{|A|} \cdot \mathtt{avg}(Y, Z) = H_{|A|}\mathtt{sep_{min}}(\mathcal{A}^k)$. Analogously, we have $\mathtt{avg}(p', A') \leq H_{|A'|}\mathtt{sep_{min}}(\mathcal{A}^k)$. Thus,

$$\mathtt{dist}(p, p') \leq H_{|A|}\mathtt{sep_{min}}(\mathcal{A}^k) + \mathtt{avg}(A, S) + \mathtt{avg}(S, A') + H_{|A'|}\mathtt{sep_{min}}(\mathcal{A}^k).$$

By averaging over all possible $S \in \mathcal{A}^k - \{A, A'\}$ we get that

$$\mathtt{dist}(p, p') \leq \cdot 2H_n\mathtt{sep_{min}}(\mathcal{A}^k) + \frac{1}{k-2}\sum_{S\notin\{A,A'\}}(\mathtt{avg}(A, S) + \mathtt{avg}(S, A'))$$

By adding over all points $p \in P \cap A$ and $p' \in P \cap A'$ we get that

$$\sum_{p \in P \cap A} \sum_{p' \in A' \cap Y} \texttt{dist}(p, p') \leq$$

$$\sum_{p \in P \cap A} \sum_{p' \in P \cap A'} 2H_n \texttt{sep}_{\min}(\mathcal{A}^k) + \frac{|P \cap A| \cdot |P \cap A'|}{k - 2} \sum_{S \notin \{A, A'\}} (\texttt{avg}(A, S) + \texttt{avg}(S, A'))$$

By adding the above inequalities for $p, p' \in P$, with $h(p) \neq h(p')$, we get that

$$\sum_{\substack{p, p' \in P \\ h(p) \neq h(p')}} \texttt{dist}(p, p') \leq \tag{16}$$

$$\sum_{\substack{p, p' \in P \\ h(p) \neq h(p')}} 2H_n \cdot \texttt{sep}_{\min}(\mathcal{A}^k) + \frac{1}{k-2} \sum_{\substack{A, A' \in \mathcal{A}^k \\ A \neq A'}} |P \cap A| \cdot |P \cap A'| \sum_{S \notin \{A, A'\}} (\texttt{avg}(A, S) + \texttt{avg}(S, A')) = \tag{17}$$

$$\sum_{\substack{p, p' \in P \\ h(p) \neq h(p')}} 2H_n \cdot \texttt{sep}_{\min}(\mathcal{A}^k) + \frac{1}{k-2} \sum_{\substack{A, A' \in \mathcal{A}^k \\ A \neq A'}} (|P \cap (A \cup A')|) \cdot (k - |P \cap (A \cup A')|) \cdot \texttt{avg}(A, A') \leq \tag{18}$$

$$\sum_{\substack{p, p' \in P \\ h(p) \neq h(p')}} 2H_n \cdot \texttt{sep}_{\min}(\mathcal{A}^k) + k \sum_{\substack{A, A' \in \mathcal{A}^k \\ A \neq A'}} \texttt{avg}(A, A'), \tag{19}$$

where the last inequality holds because $(|P \cap (A \cup A')|) \cdot (k - |P \cap (A \cup A')|) \leq k^2/4$.

If we compare inequalities (16)-(19) with inequality (15), we conclude that (16)-(19) also hold for the subscase $k = 2$.

Then, by adding inequality (14) with the inequalities (16)-(19) and also using the fact $\texttt{sep}_{\min}(\mathcal{A}^k) \leq \texttt{sep}_{\text{av}}(\mathcal{A}^k)$, we get that

$$\sum_{\substack{p, p' \in P \\ p \neq p'}} \texttt{dist}(p, p') \leq 2H_n \frac{k(k-1)}{2} \texttt{sep}_{\min}(\mathcal{A}^k) + k \sum_{\substack{A, A' \in \mathcal{A}^k \\ A \neq A'}} \texttt{avg}(A, A') \leq (2H_n + k) \frac{k(k-1)\texttt{sep}_{\text{av}}(\mathcal{A}^k)}{2}$$

Proposition 4.1 ensures that

$$\frac{k(k-1)}{2} \text{OPT}_{\text{SEP}}(k) \leq \frac{k(k-1)}{2} \texttt{avg}(P) = \sum_{p, p' \in P} \texttt{dist}(p, p')$$

Thus, from the two previous inequalities, we conclude that

$$\texttt{sep}_{\text{av}}(\mathcal{A}^k) \geq \frac{\text{OPT}_{\text{SEP}}(k)}{2H_n + k}.$$

$\square$

### D.3   The $\texttt{sep}_{\text{av}}$ criterion for other linkage methods

The following instances show that the separability of both `single-linkage` and `complete-linkage` can be much lower than $\frac{\text{OPT}_{\text{SEP}}(k)}{\log n}$.

For `single-linkage`, consider the instance $\mathcal{X} = A \cup B \cup \{p\}$, where $A$ contains $n - 1 - \sqrt{n}$ points and $B$ contains $\sqrt{n}$ points $b_1, \ldots, b_{\sqrt{n}}$. Moreover, we have $\texttt{dist}(x, y) = \epsilon$, for $x, y \in A$, $\texttt{dist}(b_i, x) = i$ for every point $x \in A$ and $\texttt{dist}(b_i, b_j) = |i - j|$. Moreover, $\texttt{dist}(p, x) = 1 + \epsilon$, for every point $x \in A$. and $\texttt{dist}(p, b_i) = 1 + \epsilon + i$ In this case, `single-linkage` builds the clustering $(A \cup B, \{p\})$. We have that $\texttt{sep}_{\text{av}}(A \cup B, p) \leq 2$, while $\texttt{sep}_{\text{av}}(A \cup p, B)$ is $\Omega(\sqrt{n})$.

Regarding `complete-linkage`, we consider the instance presented at Section B.2, but without the set $E$, that is, the set of points is $\mathcal{X} = A \cup B \cup C \cup D$. When $k = 2$, `complete-linkage` builds the clustering $(A \cup C, B \cup D)$ that has $\text{sep}_\text{av}$ $O(1)$ while the clustering $(A, C \cup D \cup B)$ satisfies

$$\text{sep}_\text{av}(A, C \cup D \cup B) \geq \frac{\frac{t^2}{2}(2t^2 + t)}{t(2t^2 + t))} = \frac{t}{2}.$$

Since $t = \Theta(\sqrt{n})$, we conclude that the separability of `complete-linkage` for this instance is $O(\frac{\text{OPT}_\text{SEP}(k)}{\sqrt{n}})$.

### D.4 Separability and cohesion can be conflicting

Recall that for instance $\mathcal{I}_k^{sep}$ `average-link` builds the $k$-clustering $\mathcal{A}^k = (S_1, S_2, s_1, s_2, \ldots, s_{k-2})$. Note that $\text{max-diam}(\mathcal{A}^k) = \text{max-avg}(\mathcal{A}^k) = \epsilon$. Let $\mathcal{A}'$ be a $k$-clustering different from $\mathcal{A}^k$. We argue that $\text{max-diam}(\mathcal{A}') \geq 1$ and $\text{max-avg}(\mathcal{A}')$ is $\Omega(1/k^2)$. In fact, if $\mathcal{A}'$ has a cluster $A$ that satisfies $|A| \geq 2$ and $|A \cap S_3| \geq 1$, then $\text{max-diam}(\mathcal{A}') \geq 1$ and $\text{max-avg}(\mathcal{A}')$ is $\Omega(1/k^2)$. Otherwise, if $\mathcal{A}'$ does not have such a cluster, then all points in $S_3$ must be singletons in $\mathcal{A}'$. Since $\mathcal{A}' \neq \mathcal{A}^k$, there is a cluster in $\mathcal{A}'$ that contains both a point in $S_1$ and a point in $S_2$. Thus, $\text{max-diam}(\mathcal{A}') = D$ and $\text{max-avg}(\mathcal{A}')$ is $\Omega(D/k^2)$.

Let $\mathcal{M}$ be the class of methods with bounded approximation regarding `max-diam` or to `max-avg`. Then any method $M \in \mathcal{M}$ builds the clustering $\mathcal{A}^k$. Since $\text{sep}_\text{av}(\mathcal{A}^k)$ is $O(D/k)$ and there is a $k$-clustering for $\mathcal{I}_k^{sep}$ whose $\text{sep}_\text{av}$ is $\Omega(D)$, we conclude that the approximation factor of any method $M \in \mathcal{M}$ regarding $\text{sep}_\text{av}$ is $O(1/k)$.

Now, we consider $\text{sep}_\text{min}$. We have that $\text{sep}_\text{min}(\mathcal{A}^k) = 1$. Let $\mathcal{B} = (B_1, \ldots, B_k)$ be a $k$-clustering with the following properties: (i) $|B_i \cap S_3| \geq 1$ for each $i \leq k - 2$; (ii) each $B_i$, with $i \geq 2$, has exactly one point in $S_1 \cup S_2$ (iii) $B_{k-1}$ has a point in $S_1$ and $B_k$ has a point in $S_2$. We have that $\text{sep}_\text{min}(\mathcal{B})$ is $\Omega(D)$. Thus, any method $M \in \mathcal{M}$ has approximation $O(1/D)$ to $\text{sep}_\text{min}$, that is, the approximation is unbounded in terms of $n$.

## E Proof of Theorem 5.4

*Proof.* Let $\mathcal{I}$ be the instance obtained by augmenting the instance $\mathcal{I}^{CS}$, presented right after Theorem 3.4, with the points $x_0, \ldots, x_{t-1}$, where $\text{dist}(x_i, A_i) = t + 1 + \epsilon$ and for $i \neq j$, $\text{dist}(x_i, x_j) = |p_j - p_i| + 2(t + 1 + \epsilon)$ and $\text{dist}(x_i, A_j) = |p_j - p_i| + t + 1 + \epsilon$.

Consider $t = k$. We argue that the $(k + 1)$-clustering obtained by `average-link` for $\mathcal{I}$ consists of the clusters $(B_{k-1}, \{x_0\}, \ldots, \{x_{k-1}\})$. In fact, in its first steps `average-link` obtains the $2k$-clustering $(A_0, \ldots, A_{k-1}, x_0, \ldots, x_{k-1})$ since the distance between points in $A_i$ is 0. In the next $k - 1$ steps, `average-link` does not make a merge involving a point $x_i$ because the average distance of $x_i$ to any other cluster is larger $k + 1$ and, by Lemma 3.5, the average distance between $B_{i-2}$ and $A_i$ is $i + 1 \leq k + 1$. Thus, the execution of `average-link` for $\mathcal{I}$ merges the same clusters that are merged in the instance $\mathcal{I}^{CS}$ and, then, ends up with the $(k+1)$-clustering $(B_{k-1}, \{x_0\}, \ldots, \{x_{k-1}\})$.

Thus, for instance $\mathcal{I}$, the maximum diameter of a cluster in $\mathcal{A}^k$ is at least $\text{diam}(B_{k-1})$, which is $\Omega(k^2)$, while the $k$-clustering $(x_0 \cup A_0, \ldots, x_{k-1} \cup A_{k-1})$ has diameter $k + \epsilon$. □

## F Experiments: extra details

Table 2 presents our datasets with their main characteristics.

Figures (1)-(6) show the results obtained by `single-linkage`, `complete-linkage` and `average-link`, for all datasets and the different criteria considered in the paper. For a given dataset $D$, method $M$ and criterion $\alpha$, the height of the bar is given by the average of $m_k$ for every $k$ considered in our experiments, where $m_k$ is the ratio between the value of criterion $\alpha$ achieved by method $M$ on dataset $D$ divided by the best value for criterion $\alpha$, among those achieved by `single-linkage`, `average-link` and `complete-linkage`on dataset $D$.

Table 2: Datasets

| Dataset | $n$ | $d$ | Source |
|---|---|---|---|
| Airfoil | 1501 | 5 | Brooks and Marcolini [2014] |
| Banknote | 1371 | 5 | Lohweg [2013] |
| Collins | 1000 | 19 | OpenML |
| Concrete | 1028 | 8 | Yeh [2007] |
| Digits | 1797 | 64 | Alpaydin [1998] |
| Geographical Music | 1057 | 116 | Zhou [2014] |
| Mice | 552 | 77 | Higuera and Cios [2015] |
| Qsarfish | 906 | 10 | Ballabio and Todeschini [2019] |
| Tripdvisor | 979 | 10 | Renjith [2018] |
| Vowel | 990 | 10 | UCI |

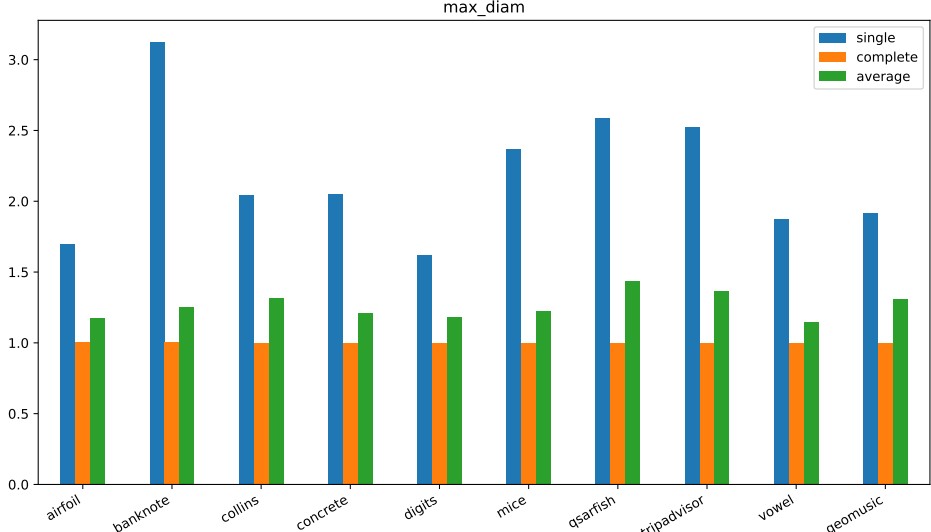

Figure 1: Results for the `max-diam` for the different datasets. For interpreting the bars, the lower the better

Regarding the cohesion criteria `complete-linkage` presents the best results for `max-diam`, followed by `average-link`. For `max-avg`, again `complete-linkage` and `average-link` are the best, with the latter having a slight advantage.

In terms of the separability criteria, `average-link` is much better than the other methods for $\text{sep}_{\min}$, while for $\text{sep}_{\text{av}}$ there is a balance between `average-link` and `single-linkage`.

For the criteria that combine cohesion and separability, `average-link` is superior for $\text{cs-ratio}_{\text{AV}}$, while there is a balance between `average-link` and `complete-linkage` for $\text{cs-ratio}_{\text{DM}}$.

Table 3 and 4 show the results for the experiment described in Section 6, when the Euclidean distance is replaced with the $\ell_1$ and $\ell_\infty$ norm, respectively. The observations made in Section 6 also hold when these metrics are used.

Finally, we note that the variance of the results for `average-link` is small. Indeed, an entry (average) close to 1 (e.g. 0.96) cannot have an underlying large variance because 1 is the maximum possible value for an entry. Since most entries for `average-link` are close to 1, one can conclude that the variance of its results is usually small. In the supplemental material, we have .csv files with our full results.

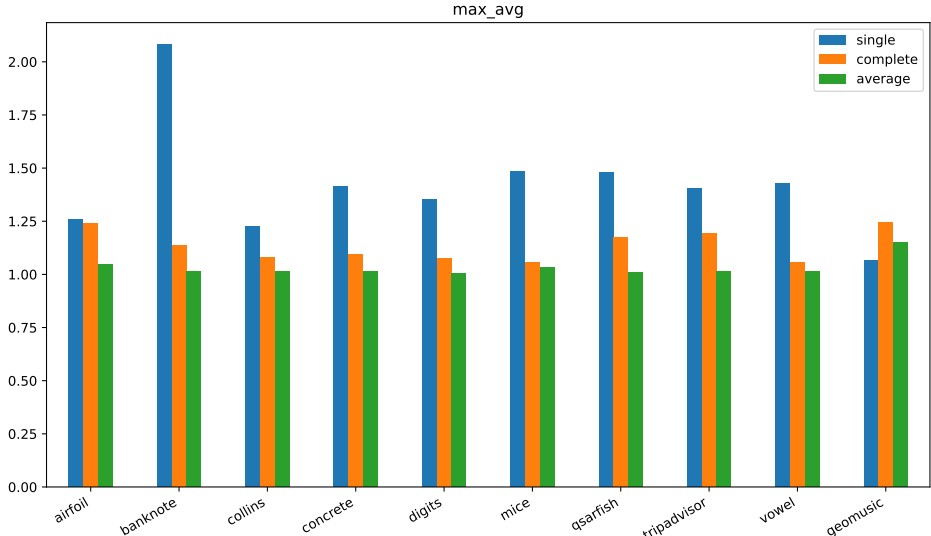

Figure 2: Results for the `max-avg` for the different datasets. For interpreting the bars, the lower the better

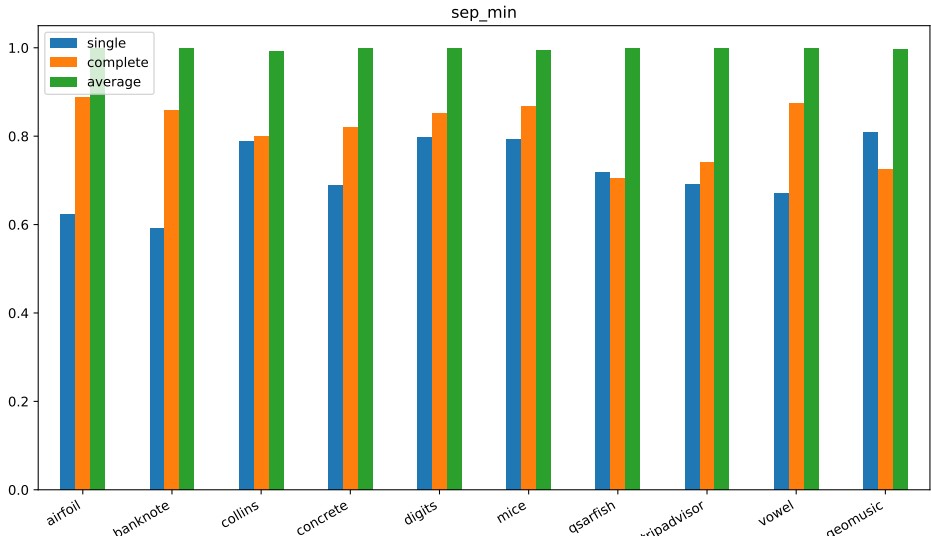

Figure 3: Results for the $\text{sep}_{\min}$ for the different datasets. For interpreting the bars, the higher the better


Figure 4: Results for the $\mathtt{sep_{av}}$ for the different datasets. For interpreting the bars, the higher the better

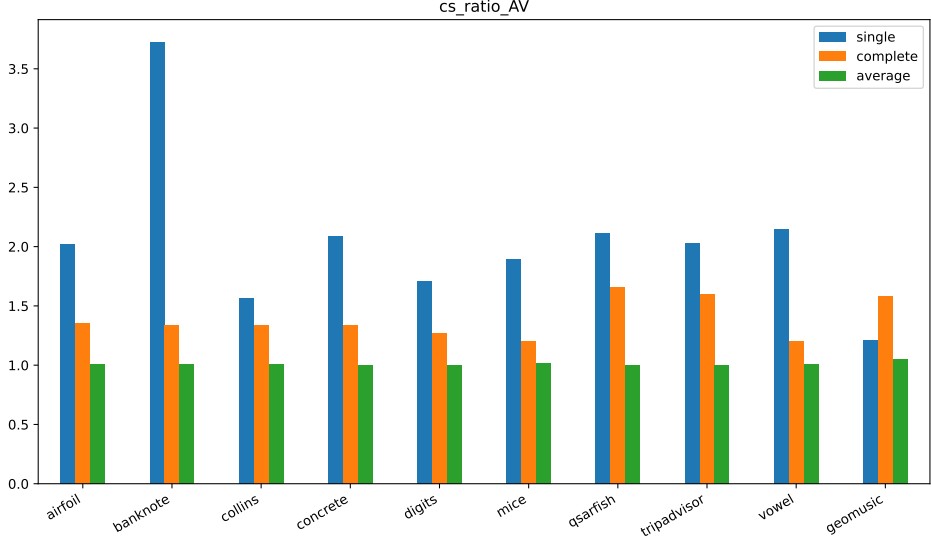

Figure 5: Results for the $\mathtt{cs\text{-}ratio_{AV}}$ for the different datasets and methods. For interpreting the bars, the lower the better

- The answer NA means that the abstract and introduction do not include the claims made in the paper.

- The abstract and/or introduction should clearly state the claims made, including the contributions made in the paper and important assumptions and limitations. A No or NA answer to this question will not be perceived well by the reviewers.

- The claims made should match theoretical and experimental results, and reflect how much the results can be expected to generalize to other settings.

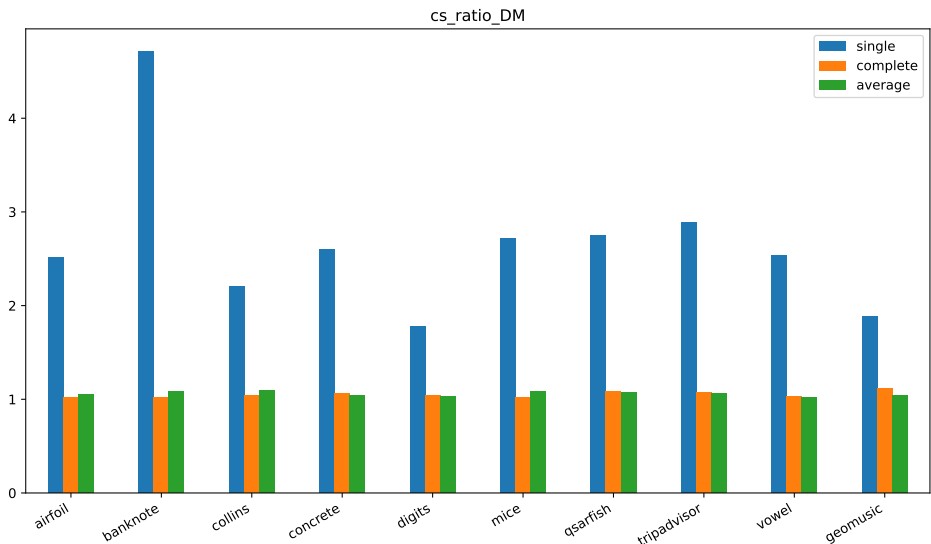

Figure 6: Results for the `cs-ratio`$_{\text{DM}}$ for the different datasets and methods. For interpreting the bars, the lower the better

Table 3: Average ratio between the result of a method and the best one for each criterion and each group of $k$. The best results are bold-faced. Distances are computed using $\ell_1$ norm

|  | Smal | | | Medium | | | Large | | |
|---|---|---|---|---|---|---|---|---|---|
|  | A | C | S | A | C | S | A | C | S |
| `sep`$_{\text{min}}$ | **0,99** | 0,81 | 0,75 | **0,99** | 0,86 | 0,66 | **0,99** | 0,9 | 0,71 |
| `sep`$_{\text{av}}$ | **0,98** | 0,83 | 0,93 | **0,96** | 0,89 | **1** | 0,97 | 0,95 | **0,99** |
| `max-diam` | 0,86 | **0,99** | 0,72 | 0,85 | **1** | 0,5 | 0,81 | **1** | 0,41 |
| `max-avg` | **0,94** | **0,94** | 0,88 | **0,99** | 0,9 | 0,73 | **0,99** | 0,83 | 0,7 |
| `cs-ratio`$_{\text{DM}}$ | **0,96** | 0,91 | 0,62 | 0,96 | **0,98** | 0,38 | 0,88 | **0,99** | 0,32 |
| `cs-ratio`$_{\text{AV}}$ | **0,98** | 0,8 | 0,71 | **1** | 0,79 | 0,51 | **1** | 0,76 | 0,51 |

- It is fine to include aspirational goals as motivation as long as it is clear that these goals are not attained by the paper.

2. **Limitations**

   Question: Does the paper discuss the limitations of the work performed by the authors?

   Answer: [Yes]

   Justification: We included a section at the end of the paper.

Table 4: Average ratio between the result of a method and the best one for each criterion and each group of $k$. The best results are bold-faced. Distances are computed using $\ell_\infty$ norm

|  | Smal | | | Medium | | | Large | | |
|---|---|---|---|---|---|---|---|---|---|
|  | A | C | S | A | C | S | A | C | S |
| `sep`$_{\text{min}}$ | **0,99** | 0,82 | 0,77 | **0,98** | 0,91 | 0,7 | **0,99** | 0,94 | 0,75 |
| `sep`$_{\text{av}}$ | **0,97** | 0,82 | 0,95 | **0,97** | 0,92 | **1** | 0,98 | 0,96 | **1** |
| `max-diam` | 0,94 | **1** | 0,9 | 0,87 | **1** | 0,7 | 0,85 | **1** | 0,56 |
| `max-avg` | 0,94 | **0,96** | 0,91 | **0,94** | 0,88 | 0,79 | **0,95** | 0,85 | 0,81 |
| `cs-ratio`$_{\text{DM}}$ | **0,97** | 0,86 | 0,74 | 0,91 | **0,98** | 0,52 | 0,89 | **0,99** | 0,45 |
| `cs-ratio`$_{\text{AV}}$ | **0,96** | 0,82 | 0,74 | **0,96** | 0,85 | 0,59 | **0,97** | 0,82 | 0,65 |

Guidelines:

- The answer NA means that the paper has no limitation while the answer No means that the paper has limitations, but those are not discussed in the paper.
- The authors are encouraged to create a separate "Limitations" section in their paper.
- The paper should point out any strong assumptions and how robust the results are to violations of these assumptions (e.g., independence assumptions, noiseless settings, model well-specification, asymptotic approximations only holding locally). The authors should reflect on how these assumptions might be violated in practice and what the implications would be.
- The authors should reflect on the scope of the claims made, e.g., if the approach was only tested on a few datasets or with a few runs. In general, empirical results often depend on implicit assumptions, which should be articulated.
- The authors should reflect on the factors that influence the performance of the approach. For example, a facial recognition algorithm may perform poorly when image resolution is low or images are taken in low lighting. Or a speech-to-text system might not be used reliably to provide closed captions for online lectures because it fails to handle technical jargon.
- The authors should discuss the computational efficiency of the proposed algorithms and how they scale with dataset size.
- If applicable, the authors should discuss possible limitations of their approach to address problems of privacy and fairness.
- While the authors might fear that complete honesty about limitations might be used by reviewers as grounds for rejection, a worse outcome might be that reviewers discover limitations that aren't acknowledged in the paper. The authors should use their best judgment and recognize that individual actions in favor of transparency play an important role in developing norms that preserve the integrity of the community. Reviewers will be specifically instructed to not penalize honesty concerning limitations.

3. **Theory Assumptions and Proofs**

   Question: For each theoretical result, does the paper provide the full set of assumptions and a complete (and correct) proof?

   Answer: [Yes]

   Justification:

   Guidelines:

   - The answer NA means that the paper does not include theoretical results.
   - All the theorems, formulas, and proofs in the paper should be numbered and cross-referenced.
   - All assumptions should be clearly stated or referenced in the statement of any theorems.
   - The proofs can either appear in the main paper or the supplemental material, but if they appear in the supplemental material, the authors are encouraged to provide a short proof sketch to provide intuition.
   - Inversely, any informal proof provided in the core of the paper should be complemented by formal proofs provided in appendix or supplemental material.
   - Theorems and Lemmas that the proof relies upon should be properly referenced.

4. **Experimental Result Reproducibility**

   Question: Does the paper fully disclose all the information needed to reproduce the main experimental results of the paper to the extent that it affects the main claims and/or conclusions of the paper (regardless of whether the code and data are provided or not)?

   Answer: [Yes]

   Justification:

   Guidelines:

   - The answer NA means that the paper does not include experiments.

- If the paper includes experiments, a No answer to this question will not be perceived well by the reviewers: Making the paper reproducible is important, regardless of whether the code and data are provided or not.
- If the contribution is a dataset and/or model, the authors should describe the steps taken to make their results reproducible or verifiable.
- Depending on the contribution, reproducibility can be accomplished in various ways. For example, if the contribution is a novel architecture, describing the architecture fully might suffice, or if the contribution is a specific model and empirical evaluation, it may be necessary to either make it possible for others to replicate the model with the same dataset, or provide access to the model. In general. releasing code and data is often one good way to accomplish this, but reproducibility can also be provided via detailed instructions for how to replicate the results, access to a hosted model (e.g., in the case of a large language model), releasing of a model checkpoint, or other means that are appropriate to the research performed.
- While NeurIPS does not require releasing code, the conference does require all submissions to provide some reasonable avenue for reproducibility, which may depend on the nature of the contribution. For example
  (a) If the contribution is primarily a new algorithm, the paper should make it clear how to reproduce that algorithm.
  (b) If the contribution is primarily a new model architecture, the paper should describe the architecture clearly and fully.
  (c) If the contribution is a new model (e.g., a large language model), then there should either be a way to access this model for reproducing the results or a way to reproduce the model (e.g., with an open-source dataset or instructions for how to construct the dataset).
  (d) We recognize that reproducibility may be tricky in some cases, in which case authors are welcome to describe the particular way they provide for reproducibility. In the case of closed-source models, it may be that access to the model is limited in some way (e.g., to registered users), but it should be possible for other researchers to have some path to reproducing or verifying the results.

5. **Open access to data and code**

   Question: Does the paper provide open access to the data and code, with sufficient instructions to faithfully reproduce the main experimental results, as described in supplemental material?

   Answer: [Yes]

   Justification:

   Guidelines:

   - The answer NA means that paper does not include experiments requiring code.
   - Please see the NeurIPS code and data submission guidelines (`https://nips.cc/public/guides/CodeSubmissionPolicy`) for more details.
   - While we encourage the release of code and data, we understand that this might not be possible, so "No" is an acceptable answer. Papers cannot be rejected simply for not including code, unless this is central to the contribution (e.g., for a new open-source benchmark).
   - The instructions should contain the exact command and environment needed to run to reproduce the results. See the NeurIPS code and data submission guidelines (`https://nips.cc/public/guides/CodeSubmissionPolicy`) for more details.
   - The authors should provide instructions on data access and preparation, including how to access the raw data, preprocessed data, intermediate data, and generated data, etc.
   - The authors should provide scripts to reproduce all experimental results for the new proposed method and baselines. If only a subset of experiments are reproducible, they should state which ones are omitted from the script and why.
   - At submission time, to preserve anonymity, the authors should release anonymized versions (if applicable).

- Providing as much information as possible in the supplemental material (appended to the paper) is recommended, but including URLs to data and code is permitted.

6. **Experimental Setting/Details**

   Question: Does the paper specify all the training and test details (e.g., data splits, hyperparameters, how they were chosen, type of optimizer, etc.) necessary to understand the results?

   Answer: [Yes]

   Justification: The details are in the paper and also in the supplemental material.

   Guidelines:

   - The answer NA means that the paper does not include experiments.
   - The experimental setting should be presented in the core of the paper to a level of detail that is necessary to appreciate the results and make sense of them.
   - The full details can be provided either with the code, in appendix, or as supplemental material.

7. **Experiment Statistical Significance**

   Question: Does the paper report error bars suitably and correctly defined or other appropriate information about the statistical significance of the experiments?

   Answer: [No]

   Justification: We have not included error bars because they do not help much in our case. However, from the tables and our analyses, the reader should have a clear idea of the variability of our results (see last paragraph of Section F).

   Guidelines:

   - The answer NA means that the paper does not include experiments.
   - The authors should answer "Yes" if the results are accompanied by error bars, confidence intervals, or statistical significance tests, at least for the experiments that support the main claims of the paper.
   - The factors of variability that the error bars are capturing should be clearly stated (for example, train/test split, initialization, random drawing of some parameter, or overall run with given experimental conditions).
   - The method for calculating the error bars should be explained (closed form formula, call to a library function, bootstrap, etc.)
   - The assumptions made should be given (e.g., Normally distributed errors).
   - It should be clear whether the error bar is the standard deviation or the standard error of the mean.
   - It is OK to report 1-sigma error bars, but one should state it. The authors should preferably report a 2-sigma error bar than state that they have a 96% CI, if the hypothesis of Normality of errors is not verified.
   - For asymmetric distributions, the authors should be careful not to show in tables or figures symmetric error bars that would yield results that are out of range (e.g. negative error rates).
   - If error bars are reported in tables or plots, The authors should explain in the text how they were calculated and reference the corresponding figures or tables in the text.

8. **Experiments Compute Resources**

   Question: For each experiment, does the paper provide sufficient information on the computer resources (type of compute workers, memory, time of execution) needed to reproduce the experiments?

   Answer: [No]

   Justification: This information is irrelevant to reproducing our experiments or reaching our conclusions.

   Guidelines:

   - The answer NA means that the paper does not include experiments.

- The paper should indicate the type of compute workers CPU or GPU, internal cluster, or cloud provider, including relevant memory and storage.
- The paper should provide the amount of compute required for each of the individual experimental runs as well as estimate the total compute.
- The paper should disclose whether the full research project required more compute than the experiments reported in the paper (e.g., preliminary or failed experiments that didn't make it into the paper).

9. **Code Of Ethics**

Question: Does the research conducted in the paper conform, in every respect, with the NeurIPS Code of Ethics `https://neurips.cc/public/EthicsGuidelines`?

Answer: [Yes]

Justification:

Guidelines:

- The answer NA means that the authors have not reviewed the NeurIPS Code of Ethics.
- If the authors answer No, they should explain the special circumstances that require a deviation from the Code of Ethics.
- The authors should make sure to preserve anonymity (e.g., if there is a special consideration due to laws or regulations in their jurisdiction).

10. **Broader Impacts**

Question: Does the paper discuss both potential positive societal impacts and negative societal impacts of the work performed?

Answer: [NA]

Justification: Our paper is mostly about theoretical results. We provide several new analyses for algorithms that are widely known. We do not see a clear societal impact that deserves to be mentioned.

Guidelines:

- The answer NA means that there is no societal impact of the work performed.
- If the authors answer NA or No, they should explain why their work has no societal impact or why the paper does not address societal impact.
- Examples of negative societal impacts include potential malicious or unintended uses (e.g., disinformation, generating fake profiles, surveillance), fairness considerations (e.g., deployment of technologies that could make decisions that unfairly impact specific groups), privacy considerations, and security considerations.
- The conference expects that many papers will be foundational research and not tied to particular applications, let alone deployments. However, if there is a direct path to any negative applications, the authors should point it out. For example, it is legitimate to point out that an improvement in the quality of generative models could be used to generate deepfakes for disinformation. On the other hand, it is not needed to point out that a generic algorithm for optimizing neural networks could enable people to train models that generate Deepfakes faster.
- The authors should consider possible harms that could arise when the technology is being used as intended and functioning correctly, harms that could arise when the technology is being used as intended but gives incorrect results, and harms following from (intentional or unintentional) misuse of the technology.
- If there are negative societal impacts, the authors could also discuss possible mitigation strategies (e.g., gated release of models, providing defenses in addition to attacks, mechanisms for monitoring misuse, mechanisms to monitor how a system learns from feedback over time, improving the efficiency and accessibility of ML).

11. **Safeguards**

Question: Does the paper describe safeguards that have been put in place for responsible release of data or models that have a high risk for misuse (e.g., pretrained language models, image generators, or scraped datasets)?

Answer: [NA]

Justification:

Guidelines:

- The answer NA means that the paper poses no such risks.
- Released models that have a high risk for misuse or dual-use should be released with necessary safeguards to allow for controlled use of the model, for example by requiring that users adhere to usage guidelines or restrictions to access the model or implementing safety filters.
- Datasets that have been scraped from the Internet could pose safety risks. The authors should describe how they avoided releasing unsafe images.
- We recognize that providing effective safeguards is challenging, and many papers do not require this, but we encourage authors to take this into account and make a best faith effort.

12. **Licenses for existing assets**

Question: Are the creators or original owners of assets (e.g., code, data, models), used in the paper, properly credited and are the license and terms of use explicitly mentioned and properly respected?

Answer: [Yes]

Justification: We cite the datasets we use in Appendix F

Guidelines:

- The answer NA means that the paper does not use existing assets.
- The authors should cite the original paper that produced the code package or dataset.
- The authors should state which version of the asset is used and, if possible, include a URL.
- The name of the license (e.g., CC-BY 4.0) should be included for each asset.
- For scraped data from a particular source (e.g., website), the copyright and terms of service of that source should be provided.
- If assets are released, the license, copyright information, and terms of use in the package should be provided. For popular datasets, `paperswithcode.com/datasets` has curated licenses for some datasets. Their licensing guide can help determine the license of a dataset.
- For existing datasets that are re-packaged, both the original license and the license of the derived asset (if it has changed) should be provided.
- If this information is not available online, the authors are encouraged to reach out to the asset's creators.

13. **New Assets**

Question: Are new assets introduced in the paper well documented and is the documentation provided alongside the assets?

Answer: [Yes]

Justification: Our supplementary material contains our codes.

Guidelines:

- The answer NA means that the paper does not release new assets.
- Researchers should communicate the details of the dataset/code/model as part of their submissions via structured templates. This includes details about training, license, limitations, etc.
- The paper should discuss whether and how consent was obtained from people whose asset is used.
- At submission time, remember to anonymize your assets (if applicable). You can either create an anonymized URL or include an anonymized zip file.

14. **Crowdsourcing and Research with Human Subjects**

Question: For crowdsourcing experiments and research with human subjects, does the paper include the full text of instructions given to participants and screenshots, if applicable, as well as details about compensation (if any)?

Answer: [NA]

Justification:

Guidelines:

- The answer NA means that the paper does not involve crowdsourcing nor research with human subjects.
- Including this information in the supplemental material is fine, but if the main contribution of the paper involves human subjects, then as much detail as possible should be included in the main paper.
- According to the NeurIPS Code of Ethics, workers involved in data collection, curation, or other labor should be paid at least the minimum wage in the country of the data collector.

15. **Institutional Review Board (IRB) Approvals or Equivalent for Research with Human Subjects**

Question: Does the paper describe potential risks incurred by study participants, whether such risks were disclosed to the subjects, and whether Institutional Review Board (IRB) approvals (or an equivalent approval/review based on the requirements of your country or institution) were obtained?

Answer: [NA]

Justification:

Guidelines:

- The answer NA means that the paper does not involve crowdsourcing nor research with human subjects.
- Depending on the country in which research is conducted, IRB approval (or equivalent) may be required for any human subjects research. If you obtained IRB approval, you should clearly state this in the paper.
- We recognize that the procedures for this may vary significantly between institutions and locations, and we expect authors to adhere to the NeurIPS Code of Ethics and the guidelines for their institution.
- For initial submissions, do not include any information that would break anonymity (if applicable), such as the institution conducting the review.

