# OpenReview forum: "On the cohesion and separability of average-link for hierarchical agglomerative clustering"
_NeurIPS.cc/2024/Conference — NeurIPS 2024 poster_

### Official Review · Reviewer_QwrC · 2024-07-03

**Soundness:** 4
**Presentation:** 3
**Contribution:** 3
**Rating:** 6
**Confidence:** 3

**Summary:**

The authors analyse the theoretical properties of the so-called average-link approach for clustering points in metric spaces. They formulate cohesion and separability criteria that capture the goodness of a clustering, essentially formalising the intuition that good clusters should be densely packed and well-separated. They prove previously unknown bounds on the cohesion and separability of clusters obtained through average-link.

**Strengths:**

- The authors analyse the properties of average-link rigorously and provide mathematical proofs for their claims.
- The work establishes previously unknown bounds on the quality of clusterings obtained through average-link with respect to cohesion and separability.

**Weaknesses:**

- The work is technically involved and therefore difficult to understand in detail for readers not trained in mathematics (such as myself). I would have welcomed a less technical summary of the paper's key points.
- The paper only mentions complete-linkage and single-linkage as alternative linkage methods for clustering. A brief discussion of other clustering methods and how they compare to average-link would have been useful. Specifically, the paper suggests using average-link rather than complete-linkage or single-linkage when cohesion and separability are relevant but does not relate average-link's performance to other methods with respect to cohesion and separability.
- The empirical results are somewhat intransparent since averaging has been done across datasets. It would be easier for the reader to examine the results if they were also presented on a per-dataset basis (for example in the appendix).

**Questions:**

- The results on empirical datasets in tables 1, 3, and 4 appear to not fully agree with the theoretical results, specifically, complete-linkage and single-linkage out-perform average-link in some cases, suggesting that average-link should not always be the preferred choice. How can this be explained?
- The second-last sentence in the introduction mentions that Dasgupta's function does not reveal how good the clusters for a specific range of k are. How do the measures introduced in the present work address this shortcoming?
- The conclusions state that average-link is a better choice than complete-linkage and single-linkage when cohesion and separability are important. Are there cases when average-link should *not* be preferred?
- Dasgupta's cost function seems to accommodate the case where similarities are asymmetric whereas the current work assumes metric spaces. Could the analyses and bounds be extended to cases of asymmetric similarities between points?

Minor points:
- The heading "Smal" in the tables is missing an l.
- Are the labels for tables 3 and 4 in Appendix F placed before their captions in the LaTeX code? This would explain why they are referred to as "F and F" in line 655.

**Limitations:**

Limitations are briefly discussed at the end of the main text.

---

> ### Author Rebuttal · Authors · 2024-07-31
>
> Thanks for revising the paper and for the positive evaluation!
>
> **Issue 2**  *The paper only mentions complete-linkage and single-linkage as alternative linkage methods for clustering. A brief discussion of other clustering methods and how they compare to average-link would have been useful. Specifically, the paper suggests using average-link rather than complete-linkage or single-linkage when cohesion and separability are relevant but does not relate average-link's performance to other methods with respect to cohesion and separability.*
>
> Reply: It is not clear to us which other clustering methods we should compare with since there are many options. We compared with single-linkage and complete-linkage because they are also quite popular and closely related to average-link. Average-link has been used for a long time, and here we provide some theoretical foundation for its superior performance compared to other popular linkage methods.
>
>
> **Issue 3** The empirical results are somewhat intransparent since averaging has been done across datasets. It would be easier for the reader to examine the results if they were also presented on a per-dataset basis (for example in the appendix)
>
> Reply:  We will add results per dataset to the appendix in our revised version. Moreover, we submitted a one-page PDF with results per-dataset in the rebuttal for all referees.
>
> **Question 1**  *The results on empirical datasets in tables 1, 3, and 4 appear to not fully agree with the theoretical results, specifically, complete-linkage and single-linkage out-perform average-link in some cases, suggesting that average-link should not always be the preferred choice. How can this be explained?*
>
> Reply. In fact, for cs-ratio_dm, complete-link outperforms average-link for ranges medium and large, and for  sep_av single-link outperforms average-link for the same ranges, by a small margin. However, this is not a contradiction since the theory only guarantees the behavior for worst-case instances, it does not state that for all instances average-link outperforms single-linkage and complete-linkage.
>
> That said, the performance of average-link in our experiments is either the best or close to the best for almost all settings.
>
>
> **Question 2** *The second-last sentence in the introduction mentions that Dasgupta's function does not reveal how good the clusters for a specific range of k are. How do the measures introduced in the present work address this shortcoming?*
>
> Reply. Our measures are calculated for each k-clustering and not for the whole hierarchy, so they can depend on the number of clusters $k$. For instance, our bound (Theorem 5.3) indicates that average-link has a better approximation in terms of cohesion (max-diam) for small k than for large k.
>
>
> **Question 3**  *The conclusions state that average-link is a better choice than complete-linkage and single-linkage when cohesion and separability are important. Are there cases when average-link should not be preferred?*
>
> Reply. Yes, as an example, if one is primarily concerned about avoiding clusters that contain distant points  (minimizing max-diam) then Complete-Link seems to be a more interesting choice: the available bound for Complete-Link (Dasgupta and Laber 24) is better than that of Average-Link (Theorem 5.3) and our experiments also suggest that Complete-Link is better than  Average-Link regarding this criterion.
>
> As a second example, if one is interested in maximizing the minimum spacing between clusters (Kleinberg and Tardos, Greedy Chapter), then single-linkage gives the optimal solution, so it is the best choice.
>
>
> **Question 4**  *Dasgupta's cost function seems to accommodate the case where similarities are asymmetric whereas the current work assumes metric spaces. Could the analyses and bounds be extended to cases of asymmetric similarities between points?*
>
> Reply. Right now we do not have an answer, this could be an interesting direction for future work. Thanks for the nice question

---

> > ### Comment · Reviewer_QwrC · 2024-08-13
> >
> > Thank you for your replies! The clarifications helped me understand the paper better; I am raising my scores regarding presentation and contribution, and maintain my overall rating.

---

### Official Review · Reviewer_Sa8n · 2024-07-12

**Soundness:** 3
**Presentation:** 3
**Contribution:** 3
**Rating:** 7
**Confidence:** 2

**Summary:**

This paper studies the performance of average-link clustering in metric spaces, focusing on criteria that offer better interpretability than Dasgupta's cost function for cohesion and separability. By investigating how well average-link balances the compactness of clusters (cohesion) with the distinctiveness between clusters (separability), the analysis sheds light on the ability to produce meaningful and well-separated clusters. The paper provides instances where single-linkage and complete-linkage clustering are exponentially worse than average-link clustering with respect to average separability. It presents lower bounds on the maximum diameter of clusters generated by average-link, providing insights into the method's clustering quality and performance compared to single linkage. Experiments conducted with real datasets confirm that the theoretical results align with practical observations, suggesting that average-link performs better than other methods when both cohesion and separability are considered.

**Strengths:**

+ The paper tackles an interesting topic

+ The Related Work section is extensive, and the proposed approaches are well-placed in the existing literature

+ The methodology is adequately sound and well-explained

+ The experimental setting is extensive, and results seem to demonstrate their effectiveness

**Weaknesses:**

I don't have any specific questions for the authors, as I'm mostly satisfied with the paper.

**Questions:**

Just out of curiosity, I would only like to know if there are any plans for future directions of the work, as they were not indicated in the paper.

**Limitations:**

I don't see any additional limitations (apart from what is already mentioned), and I'm mostly satisfied with the paper.

---

> ### Author Rebuttal · Authors · 2024-07-31
>
> Thanks again for revising our paper and for your positive evaluation!
>
> **Question** *Just out of curiosity, I would only like to know if there are any plans for future directions of the work, as they were not indicated in the paper*
>
> One potential direction for future work is addressing the case in which the input is given by similarities  rather than distances.

---

> > ### Comment · Reviewer_Sa8n · 2024-08-12
> >
> > I thank the authors and keep my original rating.

---

### Official Review · Reviewer_k54Y · 2024-07-13

**Soundness:** 3
**Presentation:** 3
**Contribution:** 3
**Rating:** 6
**Confidence:** 3

**Summary:**

This paper theoretically investigates the effectiveness of average linkage for hierarchical agglomerative clustering. The authors consider the setting where we are clustering in a metric space and consider well motivated definitions of separability and cohesion of clustering. The performance of average linkage and other methods wrt these settings is then considered. The authors conclude the work with an empirical analysis.

**Strengths:**

The paper presents an interesting analysis of average linkage as a hierarchical clustering method.

The results help to explain the effectiveness of the method.

Strengths include:
* Thoughtful analysis of a popular and well studied method
* Understanding of cost functions and how average linkage optimizes them
* Empirical analysis following theoretical results

**Weaknesses:**

I think the paper has several merits as listed above. Weaknesses include:
* I think the presentation of Table 6 could be clearer -- I think directly presenting the results per dataset per method would be clearer.
* Perhaps more text could be used to describe the novelty of proof techniques
* Given some of the motivations, I wonder if more of the analysis on random hierarchies should be included in the main paper.

**Questions:**

Can you say more about the relationship between your work and [Großwendt et al., 2019]?

**Limitations:**

Adequately addressed the limitations

---

> ### Author Rebuttal · Authors · 2024-07-31
>
> We are glad that you saw several merits in our submission!
>
> **Issue 1**:  *I think the presentation of Table 6 could be clearer -- I think directly presenting the results per dataset per method would be clearer*
>
> Reply.  I believe you mean Table 1. This table is the best solution we found for the following problem: how to give a reasonable overview of our empirical findings in a relatively small table? Given our focus on the theoretical results, we allocated only a small space in the main text for the experiments.
>
> That said,  we agree that it is good to have results per dataset per method. We will add them to the appendix in our revised version. Moreover, we submitted a one-page .pdf with these results in the rebuttal for all referees.
>
> **Issue 2**: *Perhaps more text could be used to describe the novelty of proof techniques*
>
> **Issue 3** *Given some of the motivations, I wonder if more of the analysis on random hierarchies should be included in the main paper.*
>
> Reply for 2 and 3. The final version allows an extra page. If the paper is accepted, we will use this page to add more discussion about our proof techniques and we will also move to this page some of the analyses of random hierarchies.
>
> **Question**: *Can you say more about the relationship between your work and [Großwendt et al., 2019]?*
>
> Reply. [Großwendt et al., 2019] study Ward’s method, which is a linkage method that at each step greedily merges the clusters that yield the minimum increment in a k-means cost function. The paper shows a lower bound on the approximation of the clustering built by Ward. Regarding upper bounds, it shows that Ward has a constant approximation for instances on the real line and instances that admit well-separated clusters. No upper bound for the general case is presented.
>
> As in our paper, the focus is on analyzing a popular linkage method. However,  [Großwendt et al., 2019] address the k-means cost function with Euclidean norm while our paper addresses other criteria (max-diam, sep_min, cs-ratio, etc) in general metric spaces.  Moreover, we have a way more comprehensive set of results and there is no (at most little) overlap of techniques.

---

### Official Review · Reviewer_f81R · 2024-07-16

**Soundness:** 3
**Presentation:** 3
**Contribution:** 3
**Rating:** 6
**Confidence:** 4

**Summary:**

This paper studies the well-known average linkage algorithm. The paper notes that average linkage has better approximation guarantees with respect to (variants of) Dasgupta's cost compared to complete and single linkage. However, in certain other settings such as metric graphs the approximation factor of average linkage does not outperform random HC trees. This paper therefore deviates from analysing Dasgupta's cost, and analyses average linkage with respect to seperability and cohesion criteria. There are also some experimental results.

**Strengths:**

1.) There are quite a few solid results regarding the approximation guarantee with respect to several cohesion and separability criteria (cs-ratio, OPT_sep, and several others), and shows that it outperforms single and complete linkage. Furthermore, several tight instances are provided as well. A lot of these results are good contributions to the HC literature, and provide a more clear theoretical picture of why average linkage performs so well.

2.) The experimental results also show fairly robustly that the theoretical guarantees can be seen in practice as well.

3.) The paper is well-written and enjoyable to read.

**Weaknesses:**

There are only (very) minor weaknesses:

1.) No new algorithmic contribution, e.g., some improvement to average linkage that could improve some of these bounds.

2.) Most results only hold for points in metric space.

3.) Font size changes from page 3 onwards (line 138)

**Questions:**

1.) Are there any downstream settings/tasks where approximation guarantees with regards to max-diam, cs-ratio etc. are used? A stronger motivation for why these objectives should be/are studied would be nice to include in the paper.

**Limitations:**

yes discussed

---

> ### Author Rebuttal · Authors · 2024-07-31
>
> We are glad you enjoyed reading our paper and found our results to be solid.
>
> **Issue** *Font size changes from page 3 onwards (line 138)*
>
> Thanks for pointing it out, we will fix it.
>
> **Question** *Are there any downstream settings/tasks where approximation guarantees with regards to max-diam, cs-ratio etc. are used? A stronger motivation for why these objectives should be/are studied would be nice to include in the paper.*
>
> Reply: The main goal of our paper is to provide a comprehensive theoretical study of average-link, a popular method for Hierarchical Clustering, for which good results are often reported in the literature. Our metrics were chosen because they are natural, allow easy interpretation, and capture cohesion and separability (or both).  For instance, we believe they are way easier to explain to students or practitioners than Dasgupta's cost function.
>
> We believe that it is reasonable to design algorithms to optimize our criteria, but we do not identify anything special about our criteria to justify a greater focus on them rather than on other available criteria that capture cohesion and separability.
>
> About downstream applications,  in facility location problems, the k-center (radius) criterion is used to ensure that every client is served by a "close" facility.  In metric spaces, the optimal maximum diameter (max-diam) and the optimal k-center differ by a factor of at most 2.  Thus, by optimizing the max-diam we are also optimizing the k-center (ignoring the factor of 2).
>
> We will add this discussion to our revised version.

---

> > ### Comment · Reviewer_f81R · 2024-08-11
> >
> > I thank the authors for the rebuttal, and I maintain my positive evaluation of the paper. I also agree that the metrics are more natural and easier to interpret than Dasgupta's cost function.

---

### Author Rebuttal · Authors · 2024-08-02

We thank all the referees for their time and valuable feedback! We are happy that all reviewers are positive about our submission and that it was recognized that our results provide a more clear theoretical picture of why the well-known average linkage method performs so well.

We are attaching a one-page PDF with results per dataset (suggested by reviewers k54Y and QwrC).
In the PDF we have six graphs, one for each of our six criteria.

For a given criterion $C$, dataset $D$, and method $M$,  the bar height summarizes the results for the different values of $k$ achieved by method $M$ on dataset $D$, regarding criterion $C$. More precisely, the bar height is obtained by taking the average of $m_k$ for every $k$ considered in our experiments, where $m_k$ is the ratio between the value of criterion $C$ achieved by method  $M$ on dataset $D$ divided by the best value of $C$  among single-link, average-link and complete-link achieved on dataset $D$.

In the two graphs at the top (criteria that should be maximized), higher values are better.  For the other graphs (criteria that should be minimized), lower values are better. One can see that average-link is either the best or close to the best for most settings.

---

### Decision · Program_Chairs · 2024-09-25

**Decision:**

Accept (poster)

**Comment:**

This paper considers hierarchical clustering under recently proposed clustering objectives.  These objectives have been been quite influential, but have caveats for certain kinds of data sets.  In particular, all algorithms approximate the objective.  This paper seeks to better understand the structure of solutions returned by algorithms that perform well.

The reviewers thought this paper was well-written and of interest to the community. Both the theory and experiments are generally of interest.